# FAIRER: Fairness as Decision Rationale Alignment

## Abstract

Deep neural networks (DNNs) have achieved remarkable accuracy, but they often suffer from fairness issues, as deep models typically show distinct accuracy differences among some specific subgroups (*e.g.*, males and females). Existing research addresses this critical issue by employing fairness-aware loss functions to constrain the last-layer outputs and directly regularize DNNs. Although the fairness of DNNs is improved, it is unclear how the trained network makes a fair prediction, which limits future fairness improvements. In this paper, we investigate fairness from the perspective of decision rationale and define *neuron parity scores* to characterize the fair decision process of networks by analyzing neuron behaviors in various subgroups. Extensive empirical studies show that the unfair issue could arise from the unaligned decision rationales of subgroups. Existing fairness regularization terms fail to achieve decision rationale alignment because they only constrain last-layer outputs while ignoring intermediate neuron alignment. To address the issue, we formulate the fairness as a new task, *i.e.*, *decision rationale alignment* that requires DNNs' neurons to have consistent responses on subgroups at both intermediate processes and the final prediction. To make this idea practical during optimization, we relax the naive objective function and propose *gradient-guided parity alignment*, which encourages gradient-weighted consistency of neurons across subgroups. Extensive experiments on a variety of datasets show that our method can improve fairness while maintaining high accuracy and outperforming other baselines by a large margin. We have released our codes at https://anonymous.4open.science/r/fairer_submission-F176/.

## 1 Introduction

In the current society, there is a desperate desire for social fairness among individuals. However, as deep learning is increasingly adopted for many applications that have brought convenience to our daily lives (He et al., 2016; Devlin et al., 2019; Deng et al., 2013), DNNs still suffer from the fairness problem and often exhibit undesirable discrimination behaviors (News, 2021; 2020). For example, for an intelligent task (*e.g.*, salary prediction), a trained DNN easily presents distinct accuracy values in different subgroups (*e.g.*, male and female). The discriminatory behaviors contradict with people's growing demand for fairness, which would cause severe social consequences. To alleviate such fairness problems, a line of mitigation strategies has been constantly proposed (Zemel et al., 2013; Sarhan et al., 2020; Wang et al., 2019).

A direct regularization method to improve fairness is to relax fairness metrics as constraints in the training process (Madras et al., 2018). This regularization method is designed to reduce the disparities between different subgroups in the training and testing data (See Fig. 1 (a) vs. (b)). Although this method easily improves the fairness of DNN models, it is still unclear how the trained network makes a fair decision. For example, we do not know *how the fairness regularization terms actually affect the final network parameters and let them make a fair prediction.* Without such an understanding, we would not know the effective direction for further fairness enhancement. Existing work does not address this question and the majority of them concentrate on the last-layer outputs (*i.e.*, predictions) while ignoring the internal process. In this work, we propose to study the fairness from the perspective of decision rationale and analyze existing fairness-regularized methods through a *decision-rationale-aware analysis* method. The term 'decision rationale' is known as the reason for making a decision and could be represented as the behaviors of neurons in a DNN (Khakzar et al.,

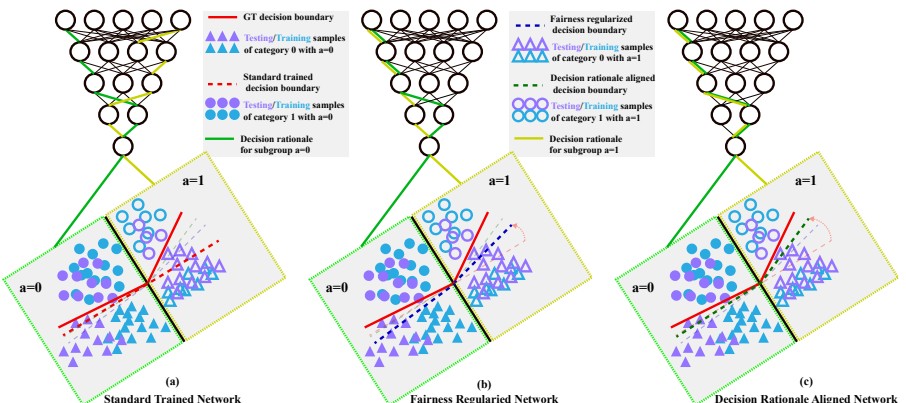

Figure 1: Schematic diagrams of two existing solutions and the proposed one. (a) and (b) represent results of the standard trained network and the regularized fairness network. (c) show the results of the decision rationale-aligned network.

2021). Specifically, for each intermediate neuron (*i.e.*, a parameter of the DNN), we can calculate the loss change on a subgroup before and after removing the neuron. As a result, we can characterize the decision rationale of a network on the subgroup by collecting the loss changes of all neurons. For example, the solid green and yellow lines in Fig. 1 represent the neurons leading to high loss changes at each layer and characterize the decision rationales of the two subgroups. Then, we define the *neuron parity score* as the decision rationale shifting across different subgroups, which actually reveals the influences of intermediate neurons (*i.e.*, parameters) to the decision rationale changes. With the new analysis tool, we find that the network fairness is directly related to the consistency of the decision rationales on different subgroups and existing fairness regularization terms could partially achieve this goal (Compare the solid lines in Fig. 1 (b)) since they only add constraints to the final outputs. Intuitively, we could define new regularization terms to minimize parity scores of all neurons and encourage them to have similar behaviors across subgroups. We name this new task as the *decision rationale alignment* that requires DNNs to have consistent decision rationales as well as final predictions on different subgroups. Although straightforward, the task is challenging for two reasons: *First*, the decision rationale and parity score are defined based on a dataset and it is impractical to calculate them at each iteration during the training process. *Second*, different neurons have different effects on fairness and such differences should be carefully considered.

To address the above two challenges, we propose the *gradient-guided parity alignment* method by relaxing the calculation of decision rationale from the dataset-based strategy to the sample-based one. As a result, the corresponding regularization term is compatible with the epoch-based training process. Moreover, we use the first-order Taylor expansion to approximate the parity score between decision rationales and the effects of different neurons to the fairness are weighted via their gradient magnitudes automatically. Overall, the proposed method can achieve much higher fairness than state-of-the-art methods. In summary, the work makes the following contributions:

1. To understand how a network makes a fair decision, we define *neuron parity score* to characterize the decision rationales of the network on different subgroups. We reveal that the fairness of a network is directly related to the consistency of its decision rationales on different subgroups and existing regularization terms cannot achieve this goal.

2. To train a fairer network, we formulate the *decision rationale alignment* task and propose the *gradient-guided parity alignment* method to solve it by addressing the complex optimization challenges.

3. Extensive experiments on three public datasets, *i.e.*, Adult, CelebA, and Credit, demonstrate that our method can enhance the fairness of DNNs effectively and outperform others largely.

## 2 RELATED WORK

**Fairness in deep learning.** There are different methods to evaluate fairness in deep learning, among which individual fairness (Zhang et al., 2020; 2021; George John et al., 2020), group fairness (Louppe et al., 2016; Moyer et al., 2018; Gupta et al., 2021; Garg et al., 2020), and counterfactual fairness (Kusner et al., 2017) are the mainstream. We focus on group fairness which is derived by calculating and comparing the predictions for each group. There is a line of work dedicated to alleviating unjustified

bias. For example, Wang et al. (2020) compare mitigation methods including oversampling, adversarial training, and other domain-independent methods. Some work proposes to disentangle unbiased representations to ensure fair DNNs. On the contrary, Du et al. (2021) directly repair the classifier head even though the middle representations are still biased. To improve fairness, it is also popular to constrain the training process by imposing regularization. Woodworth et al. (2017) regularize the covariance between predictions and sensitive attributes. Madras et al. (2018) relax the fairness metrics for optimization. Although such methods are easy to be implemented and integrated into the training process, these constraints suffer from overfitting (Cotter et al., 2019). The model with a large number of parameters could memorize the training data, which causes the fairness constraints to fit well only in the training process. Chuang & Mroueh (2021) ensure better generalization via data augmentation (*e.g.*, mix-up) to reduce the trade-off between fairness and accuracy.

However, these methods barely pay attention to the rationale behind the fair decision results. In this paper, we further analyze the decision rationales behind the fair decision results in the training process and reveal that ensuring the fair decision rationale could further improve fairness.

**Understanding DNNs decision rationale.** There are some interpretable methods enabling DNNs models to present their behaviors in understandable ways to humans (Zhang & Zhu, 2018; Fong & Vedaldi, 2017; Koh & Liang, 2017). Specifically, there is a line of work that decompose DNNs to depict the decision rationale. Routing paths composed of the critical nodes (*e.g.* neurons with the most contribution to the final classification on each layer) can be extracted in a learnable way to reflect the network's semantic information flow regarding to a group of data (Khakzar et al., 2021). Conquering the instability existing in the learnable method, Qiu et al. (2019) propose an activation based back-propagation algorithm to decompose the entire DNN model into multiple components composed of structural neurons. Meanwhile, Xie et al. (2022) base the model function analysis on the neuron contribution calculation and reveal that the neuron contribution patterns of OOD samples and adversarial samples are different from that of normal samples, resulting in wrong classification results. Zheng et al. (2022) analyze neurons sensitive to individual discrimination and generate testing cases according to sensitive neuron behaviors. However, these post-hoc methods decompose the DNNs and extract the neuron contributions pattern via static analysis or in a learnable way. These analysis methods result in huge time overhead, making their integration into the training process difficult, which restricts these methods to be applied in optimizing the training process.

In our paper, we follow the spirit of decomposing DNNs to understand the model decision rationale. Different from these previous methods, our method successfully simplifies the estimation process of neuron contribution and can be easily integrated into the training process to optimize the model.

## 3 PRELIMINARIES: GROUP FAIRNESS VIA REGULARIZATION LOSSES

### 3.1 PROBLEM FORMULATION

In general, given a dataset $\mathcal{D}$ containing data samples (*i.e.*, $\mathbf{x} \in \mathcal{X}$) and corresponding labels (*i.e.*, $y \in \mathcal{Y}$), we can train a DNN to predict the labels of input samples, *i.e.*, $\hat{y} = F(\mathbf{x})$ with $\hat{y} \in \mathcal{Y}$ being the prediction results. In the real word, the samples might be divided into subgroups according to some sensitive attributes $a \in \mathcal{A}$ such as gender and race. Without loss of generality, we consider the binary classification and binary attribute setup, *i.e.*, $y \in \{0, 1\}$ and $a \in \{0, 1\}$. For example, $a = 0$ and $a = 1$ could represent male and female, respectively. A fair DNN (*i.e.*, $F(\cdot)$) is desired to obtain a similar accuracy in the two subgroups.

### 3.2 FAIRNESS REGULARIZATION

Existing fairness works (Feldman et al., 2015; Hardt et al., 2016; Madras et al., 2018; Chuang & Mroueh, 2021) focus on designing fairness regularization terms and adding them to the loss function, which encourages the targeted DNN to predict similar results across subgroups. Specifically, Feldman et al. (2015) develop the demographic parity (DP) regularization term to encourage the predicted label to be independent of the sensitive attribute (*i.e.*, $a$), that is, $P(\hat{y}|a = 0) = P(\hat{y}|a = 1)$ which means that the probability distribution of $\hat{y}$ condition on $a = 0$ should be the same as the condition on $a = 1$. Hardt et al. (2016) further propose the equalized odds (EO) regularization to consider the ground truth label $y$ and make the prediction and sensitive attribute conditionally independent w.r.t. $y$, *i.e.*, $P(\hat{y}|a = 0, y) = P(\hat{y}|a = 1, y)$. Although straightforward, it is difficult to optimize the above

regularization terms and Madras et al. (2018) propose relaxed counterparts:

$$\Delta\text{DP}(\text{F}) = \left| \text{E}_{\mathbf{x}\sim P_0}(\text{F}(\mathbf{x})) - \text{E}_{\mathbf{x}\sim P_1}(\text{F}(\mathbf{x})) \right|, \tag{1}$$

where $P_0 = P(\mathbf{x}|a = 0)$ and $P_1 = P(\mathbf{x}|a = 1)$ are the distributions of $\mathbf{x}$ condition on $a = 0$ and $a = 1$, respectively, and the function $\text{E}(\cdot)$ is to calculate the expectation under the distributions.

$$\Delta\text{EO}(\text{F}) = \sum_{y\in\{0,1\}} \left| \text{E}_{\mathbf{x}\sim P_0^y}(\text{F}(\mathbf{x})) - \text{E}_{\mathbf{x}\sim P_1^y}(\text{F}(\mathbf{x})) \right|, \tag{2}$$

where $P_0^1 = P(\mathbf{x}|a = 0, y = 1)$ denotes the distribution of $\mathbf{x}$ condition on the $a = 0$ and $y = 1$, and we have similar notations for $P_0^0$, $P_1^1$, $P_1^0$ if we set the DNN for a binary classification task and have the label $y \in 0, 1$. We can add Eq. (1) and Eq. (2) to the classification loss (*e.g.*, cross-entropy loss) to regularize the fairness of the targeted DNN, respectively, and obtain the whole loss function

$$\mathcal{L} = \text{E}_{(\mathbf{x},y)\sim P}(\mathcal{L}_{\text{cls}}(\text{F}(\mathbf{x}), y)) + \lambda\mathcal{L}_{\text{fair}}(\text{F}), \tag{3}$$

where $P$ denotes the joint distribution of $\mathbf{x}$ and $y$, $\mathcal{L}_{\text{cls}}$ is the classification loss, and the term $\mathcal{L}_{\text{fair}}$ could be $\Delta\text{DP}(\text{F})$ or $\Delta\text{EO}(\text{F})$ defined in Eq. (1) or Eq. (2). We can minimize the above loss function and get fairness-regularized DNNs. Although effective, the above method presents some generalization limitations. To alleviate this issue, Chuang & Mroueh (2021) embed the data augmentation strategy into the fairness regularization method and propose FairMixup with novel DP- and EO-dependent regularization terms. Please refer to Chuang & Mroueh (2021) for more details. Our method is also applicable to other fairness metrics that quantify the expected difference between groups. More analysis is put in supplementary materials.

Overall, we get several fairness regularization methods via different regularization terms. Specifically, we denote the methods without augmentation as FairReg($\Delta$DP, noAug) and FairReg($\Delta$EO, noAug) based on regularization functions (*i.e.*, Eq. (1) and Eq. (2)). We denote the methods equipped with data augmentation as FairReg($\Delta$DP, Aug) and FairReg($\Delta$EO, Aug), respectively.

## 3.3 LIMITATIONS

Although the above methods are able to enhance the fairness of DNNs, they still present some limitations. We conduct an experiment on the Adult dataset (Dua & Graff, 2017a) with a neural network with 3-layer MLPs. Specifically, we train the network with two fairness regularization methods (*i.e.*, FairReg($\Delta$DP, noAug) and FairReg($\Delta$DP, Aug) [1]) and five different $\lambda \in \{0.2, 0.3, 0.4, 0.5, 0.6\}$, that is, for each method, we get five trained networks. Then, we can calculate the accuracy scores and fairness scores of all networks on the testing dataset. We employ average precision for the accuracy score and $-$DP for the fairness score since a smaller DP means better fairness. For each method, we can draw a plot

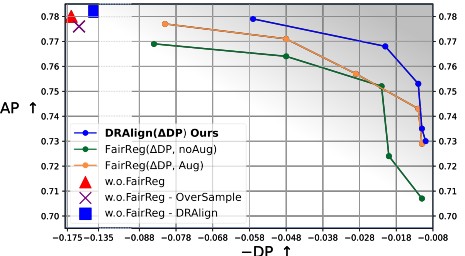

Figure 2: Accuracy and fairness comparison of five different methods on the Adult dataset. The hyperparameter $\lambda$ increases from 0.2 to 0.6 along the $-$DP axis as it becomes larger.

w.r.t. different $\lambda$. Besides, we also train a network without the fairness regularization term and denote it as w.o.FairReg. Based on w.o.FairReg, we can conduct oversampling on the training samples to balance the samples across different subgroups (Wang et al., 2020) and denote it as w.o.FairReg-Oversample. As shown in Figure 2, we see that: ❶ The standard trained network via w.o.FairReg presents an obvious fairness issue and the oversampling solution has limited capability to fix it. ❷ When we use the regularization methods and gradually increase the weight $\lambda$ in Eq. (3) from 0.2 to 0.6, FairReg($\Delta$DP, noAug) is able to generate fairer networks with higher fairness scores (*i.e.*, higher -DP) than the one from w.o.FairReg. However, the corresponding accuracy decreases by a large margin, that is, *existing methods could hardly generate enough fair networks under similar accuracy*. ❸ The data augmentation-based method (*i.e.*, FairReg($\Delta$DP, Aug)) can alleviate such an issue to some extent and achieves higher fairness than FairReg($\Delta$DP, noAug) under similar accuracy.

Such methods only provide fairness metric results but neglect the decision-making process. Although intuitively, the consistent decision process of different groups could improve the fairness performance,

---

[1]We have similar observations on the $\Delta$EO-based methods and remove them for a clear explanation.

to explore the concrete relationship between the decision-making process and fairness, we provide an analysis method by extending the decision rationale-aware explainable methods in Sec. 4. Specifically, instead of using the final fairness metrics, we define the neuron parity score for each parameter of a network that measures whether the parameter is fair, that is, whether it has consistent responses to different subgroups.

## 4 DECISION RATIONALE-AWARE FAIRNESS ANALYSIS

In recent years, decision rationale-aware explainable methods are developed and help understand how a trained network makes a decision (Khakzar et al., 2021; Wang et al., 2018). In these works, the decision rationale is represented by measuring the importance of intermediate neurons. Inspired by this idea, to understand a fair decision, we study the decision process of networks by analyzing their neuron behaviors under different subgroups and define the decision rationales for different subgroups. Then, we define the *parity score* for a network that actually measures whether the decision rationales on different subgroups are consistent. Besides, we can use the parity score to compare the networks trained with different regularization terms.

### 4.1 NEURON PARITY SCORE

Inspired by recent work on understanding the importance of the neuron for the classification loss (Molchanov et al., 2019), we define the neuron parity score based on the independent assumption across neurons (*i.e.*, parameters). When we have a trained network $F(\cdot)$ with its parameters $\mathcal{W} = \{w_0, \ldots, w_K\}$, we can calculate classification losses on samples from two distributions $P_0 = P((\mathbf{x}, y)|a = 0)$ and $P_1 = P((\mathbf{x}, y)|a = 1)$, which correspond to the training subsets of two subgroups (*i.e.*, $a = 0$ and $a = 1$) and get the losses $\mathcal{J}(F, P_0)$ and $\mathcal{J}(F, P_1)$, respectively. Meanwhile, we can modify $F(\cdot)$ by removing a specific parameter $w_k$ and denote the new counterpart as $F_{w_k=0}$, and we can also obtain losses via $\mathcal{J}(F_{w_k=0}, P_0)$ and $\mathcal{J}(F_{w_k=0}, P_1)$. Then, for each subgroup (*i.e.*, $P_0$ or $P_1$), we calculate the loss change before and after removing the parameter $w_k$ by

$$c_k^{a=i} = C(F, w_k, P_i) = |\mathcal{J}(F, P_i) - \mathcal{J}(F_{w_k=0}, P_i)|^2, \ \forall i \in \{0, 1\}, k \in [0, K], \tag{4}$$

where the function $\mathcal{J}(F, P_i)$ is to calculate the classification loss (*i.e.*, $\mathcal{L}_{cls}$ in Eq. (3)) of examples in $P_i$ with $\forall i \in \{0, 1\}$ based on the network $F$. With a subgroup $P_i$ and a $K$-neuron network $F$, we can get $\mathbf{c}_F^{a=i} = [c_0^{a=i}, c_1^{a=i}, \ldots, c_K^{a=i}]$ that is regarded as a representation of the decision rationale on the subgroup $P_i$ (Khakzar et al., 2021).

Then, we define the parity score of the parameter $w_k$ as the difference between $c_k^{a=0} = C(F, w_k, P_0)$ and $c_k^{a=1} = C(F, w_k, P_1)$, *i.e.*,

$$d_k = |C(F, w_k, P_0) - C(F, w_k, P_1)|^2. \tag{5}$$

Intuitively, if the network $F$ is fair to a kind of sensitive attribute, each parameter should have consistent responses to different subgroups, and the changes before and after removing the parameter should be the same. As a result, a smaller $d_k$ means that the parameter $w_k$ is less sensitive to the attribute changes. For the entire network with $K$ neurons, we get $K$ parity scores and $\mathbf{c}_F^{a=i} = [c_0^{a=i}, c_1^{a=i}, \ldots, c_K^{a=i}]$, and can represent the network with $\mathbf{d}_F = [d_0, d_1, \ldots, d_K]$ and aggregate all scores for a network-level parity score, *i.e.*, $d_F = \sum_{k=0}^{K} d_k = |\mathbf{c}_F^{a=0} - \mathbf{c}_F^{a=1}|_1$, which measures whether the decision rationales on the two subsets are consistent (*i.e.*, properly aligned).

### 4.2 RELATIONSHIP BETWEEN PARITY SCORE AND FAIRNESS

With the *neuron parity score*, we conduct an empirical study based on the Adult dataset and a neural network with 3-layer MLPs. Specifically, we train six networks with the regularization terms defined in Sec. 3.3, *e.g.*, the $\Delta$DP-based regularization terms with six different weights (*i.e.*, FairReg($\Delta$DP, noAug) with $\lambda \in \{0.0, 0.2, 0.3, 0.4, 0.5, 0.6\}$). Note that, FairReg($\Delta$DP, noAug) with $\lambda = 0.0$ represents the standard trained network without fair regularization terms (*i.e.*, w.o.FairReg). Then, for each method, we can train a neural network and calculate the parity score, *i.e.*, $d_F = \sum_{k=0}^{K} d_k = |\mathbf{c}_F^{a=0} - \mathbf{c}_F^{a=1}|_1$ to measure the decision rationale shifting across subgroups and the fairness score defined by $-$DP. As reported in Table 1, we see that: ❶ the parity score of the network gradually decreases as the -DP becomes higher, which demonstrates that the fairness

Table 1: Parity scores, fairness scores, and the first-order Taylor approximation of the parity scores of networks trained via FairReg($\Delta$DP, noAug) with different $\lambda$ in Eq. (3). For each network, we train 10 runs with different seeds and the average results are reported.

| | FairReg($\Delta$DP, noAug) | | | | | |
|---|---|---|---|---|---|---|
| | $\lambda = 0.0$ | $\lambda = 0.2$ | $\lambda = 0.3$ | $\lambda = 0.4$ | $\lambda = 0.5$ | $\lambda = 0.6$ |
| Parity score ($d_F$) | $0.624 \pm 0.555$ | $0.391 \pm 0.338$ | $0.101 \pm 0.084$ | $0.070 \pm 0.045$ | $0.046 \pm 0.023$ | $0.039 \pm 0.029$ |
| Fairness ($-$DP) | $-0.160 \pm 0.012$ | $-0.084 \pm 0.009$ | $-0.048 \pm 0.004$ | $-0.022 \pm 0.006$ | $-0.020 \pm 0.003$ | $-0.010 \pm 0.003$ |
| Approx. ($-\sum_{l=0}^{L} \cos(\vec{c}_l^{a=0}, \vec{c}_l^{a=1})$) | $-0.670 \pm 0.042$ | $-1.380 \pm 0.093$ | $-1.530 \pm 0.111$ | $-1.630 \pm 0.187$ | $-1.630 \pm 0.156$ | $-1.800 \pm 0.108$ |

of a network is highly related to the decision rationale shifting across subgroups. ❷ adding the fairness regularization term on the last-layer outputs (*i.e.*, $\lambda > 0$) can decrease the decision rationale shifting to some extent. However, such an indirect way could hardly achieve the optimized results and a more effective way is to actively align the decision rationale explicitly. Note that we can observe similar results on other regularization methods and focus on FairReg($\Delta$DP, noAug) due to the limited space. We conclude that the existing fairness regularization-based methods can encourage the consistency between decision rationales of the network on different subgroups to some extent and get smaller neuron parity scores than the standard trained network. This inspires our method in Sec.5 that conducts alignment of the decision rationales of different subgroups explicitly.

## 5 DECISION RATIONALE ALIGNMENT

### 5.1 FORMULATION AND CHALLENGES

According to Eq. (5), we can achieve a fairer network by aligning the decision rationales of subgroups and a straightforward way is to set the parity score $d_F = \sum_{k=0}^{K} d_k$ as an extra loss function and minimize it directly, that is, we can add a new loss to Eq. (3) and have,

$$\mathcal{L} = E_{(\mathbf{x},y)\sim P}(\mathcal{L}_{\text{cls}}(F(\mathbf{x}), y)) + \lambda \mathcal{L}_{\text{fair}}(F) + \beta \sum_{k=0}^{K} d_k, \quad (6)$$

where $d_k$ is the parity score of the $k$th neuron and calculated by Eq. (5). Such a loss should calculate parity scores for all neurons and all samples in a dataset, leading to a high cost and is not practical.

### 5.2 GRADIENT-GUIDED PARITY ALIGNMENT

To address the challenges, we relax Eq. (4) to the sample-based counterpart

$$c_k^{a=i} = C(F, w_k, P_i) = |E_{(\mathbf{x},y)\sim P_i}(\mathcal{L}_{\text{cls}}(F(\mathbf{x}), y) - E_{(\mathbf{x},y)\sim P_i}(\mathcal{L}_{\text{cls}}(F_{w_k=0}(\mathbf{x}), y))|^2, \quad (7)$$
$$\forall i \in \{0, 1\}, k \in [0, K].$$

We use the first-order Taylor expansion to approximate $c_k^{a=i}$ similar to Molchanov et al. (2019) and get

$$\hat{c}_k^{a=i} = \hat{C}(F, w_k, P_i) = (g_k^{a=i} \cdot w_k)^2, \forall i \in \{0, 1\}, k \in [0, K]. \quad (8)$$

where $g_k^{a=i}$ denotes the gradient of the $k$th neuron (*i.e.*, $w_k$) w.r.t. the loss function on the examples sampled from the distribution of the $i$th subgroup (*i.e.*, $P_i$). Intuitively, the above definition means that we should pay more attention to the neurons with higher gradients and make them have similar responses to different subgroups. However, neurons (*i.e.*, parameters) of different layers may have different score ranges. To avoid this influence, we further normalize $\hat{c}_k^{a=i}$ by $\frac{\hat{c}_k^{a=i}}{|\hat{\mathbf{c}}_l^{a=i}|} \forall i \in \{0, 1\}, k \in \mathcal{K}_l$, where $\mathcal{K}_l$ contains the indexes of the neurons in the $l$th layer, and parity scores of neurons in the $l$th layer (*i.e.*, $\{\hat{c}_k^{a=i} | k \in \mathcal{K}_l\}$) form a vector $\hat{\mathbf{c}}_l^{a=i} = \text{vec}(\{\hat{c}_k^{a=i} | k \in \mathcal{K}_l\})$. Then, we can get a new vector for the $l$th layer $\vec{c}_l^{a=i} = \text{vec}(\{\frac{\hat{c}_k^{a=i}}{|\hat{\mathbf{c}}_l^{a=i}|} | k \in \mathcal{K}_l\})$ by normalizing each element. Then, we can update Eq. (6) by minimizing the distance between $\vec{c}_l^{a=0}$ and $\vec{c}_l^{a=1}$ $\forall l \in [0, L]$, *i.e.*,

$$\mathcal{L} = E_{(\mathbf{x},y)\sim P}(\mathcal{L}_{\text{cls}}(F(\mathbf{x}), y)) + \lambda \mathcal{L}_{\text{fair}}(F) - \beta \sum_{l=0}^{L} \cos(\vec{c}_l^{a=0}, \vec{c}_l^{a=1}), \quad (9)$$

where $L$ denotes the number of layers in the network, and the function $\cos(\cdot)$ is the cosine similarity function. The last two terms are used to align the final predictions and the responses of the intermediate neurons across subgroups, respectively. To validate the approximation (*i.e.*, $-\sum_{l=0}^{L} \cos(\vec{c}_l^{a=0}, \vec{c}_l^{a=1})$) can reflect the decision rationale alignment degree like the parity score $\sum_{k=0}^{K} d_k$. We conduct an empirical study on FairReg($\Delta$DP, noAug) as done in Sec. 4.2 and calculate the value of $-\sum_{l=0}^{L} \cos(\vec{c}_l^{a=0}, \vec{c}_l^{a=1})$ for all trained networks. From Table 1, we see that the approximation has consistent variation trend with the parity score under different $\lambda$.

---

**Algorithm 1:** Gradient-guided Parity Alignment

---

**Data:** Network F with parameters $\mathcal{W} = \{w_0, \ldots, w_K\}$, epoch index set $\mathcal{E}$, training data $\mathcal{D}$, batch size $B$, network layers $L$, neurons in the $l$th layer $\mathcal{K}_l$, hyper-parameters $\lambda$ and $\beta$, learning rate $\eta$.

```
// Training process for DP
```
1 **for** $e \in \mathcal{E}$ **do**
```
     // Sampling B samples from different subgroups in D
```
2      $(\mathbf{X}_0, \mathbf{Y}_0) \leftarrow \text{Sample}(\mathcal{D}, a = 0, B); (\mathbf{X}_1, \mathbf{Y}_1) \leftarrow \text{Sample}(\mathcal{D}, a = 1, B);$
```
     // Calculating loss and updating the model
```
3      $\mathcal{L}_{\text{cls}}(\text{F}(\mathbf{X}_0), \mathbf{Y}_0), \mathcal{L}_{\text{cls}}(\text{F}(\mathbf{X}_1), \mathbf{Y}_1); \mathcal{L}_{\text{fair}} = \Delta \text{DP}(\text{F}, \mathbf{X}_0, \mathbf{X}_1);$
4      **for** $l \in L$ **do**
5          **for** $k \in \mathcal{K}_l$ **do**
6              $g_k^{a=0} = \frac{\partial(\mathcal{L}_{\text{cls}}(\text{F}(\mathbf{X_0}), \mathbf{Y_0}))}{\partial w_k}; g_k^{a=1} = \frac{\partial(\mathcal{L}_{\text{cls}}(\text{F}(\mathbf{X_1}), \mathbf{Y_1}))}{\partial w_k};$
7              $\hat{c}_k^{a=0} \leftarrow (g_k^{a=0} \cdot w_k)^2; \hat{c}_k^{a=1} \leftarrow (g_k^{a=1} \cdot w_k)^2;$
8          $\vec{\mathbf{c}}_l^{a=0} = [\hat{c}_0^{a=0}, \hat{c}_1^{a=0}, \ldots, \hat{c}_{|\mathcal{K}_l|}^{a=0}]; \vec{\mathbf{c}}_l^{a=1} = [\hat{c}_0^{a=1}, \hat{c}_1^{a=1}, \ldots, \hat{c}_{|\mathcal{K}_l|}^{a=1}];$
9      $\mathcal{L} \leftarrow \mathcal{L}_{\text{cls}}(\text{F}(\mathbf{X}_0), \mathbf{Y}_0) + \mathcal{L}_{\text{cls}}(\text{F}(\mathbf{X}_1), \mathbf{Y}_1) + \lambda \mathcal{L}_{fair} - \beta \sum_{l=0}^{L} \cos(\vec{\mathbf{c}}_l^{a=0}, \vec{\mathbf{c}}_l^{a=1});$
10      $w \leftarrow w - \eta \nabla_w \mathcal{L}, \forall w \in \mathcal{W}.$

---

### 5.3 IMPLEMENTATION DETAILS

We detail the whole training process in Algorithm 1. In particular, given a training dataset $\mathcal{D}$, we first sample two groups of samples (ie, $(\mathbf{X}_0, \mathbf{Y}_0)$ and $\mathbf{X}_1, \mathbf{Y}_1$) from the two subgroups in the dataset, respectively (See line 2). Then, we calculate the cross-entropy loss for both samples (See line 3) and calculate the fairness regularization loss (*i.e.*, $\mathcal{L}_{\text{fair}} = \Delta \text{DP}(\text{F}, \mathbf{X}_0, \mathbf{X}_1)$. After that, we can calculate the gradient of each parameter (*i.e.*, neuron $w_k$) w.r.t. the classification loss (See line 6) and calculate the decision rationale for each neuron and layer (See line 7 and 8). Finally, we calculate the cosine similarity between $\vec{\mathbf{c}}_l^{a=0}$ and $\vec{\mathbf{c}}_l^{a=1}$ and use the whole loss to update the parameters. *We defer the algorithm depiction for the EO metric to the Supplementary Materials.*

## 6 EXPERIMENTS

### 6.1 EXPERIMENTAL SETUP

**Datasets.** In our experiments, we use two tabular benchmarks (**Adult** and **Credit**) and one image dataset (**CelebA**) that are all for binary classification tasks: ❶ Adult (Dua & Graff, 2017a). The original aim of the dataset Adult is to determine whether a person makes salaries over 50K a year. We consider *gender* as the sensitive attribute, and the vanilla training will lead the model to predict females to earn less salaries. ❷ CelebA (Liu et al., 2015). The CelebFaces Attributes dataset is to predict the attributes of face. We split into two subgroups according to the attribute *gender*. Here we consider two attributes classification tasks. For the task to predict whether the hair in an image is *wavy* or not, the standard training will show discrimination towards the male group; when predicting whether the face is *attractive*, the standard training will result in a model prone to predict males as less attractive. ❸ Credit (Dua & Graff, 2017b). This dataset is to give an assessment of credit based on personal and financial records. In our paper, we take the attribute *gender* as the sensitive attribute.

**Models.** For tabular benchmarks, we use the MLP (multilayer perception) (Bishop, 1996) as the classification model, which is commonly adopted in classifying tabular data. For the CelebA dataset, we use AlexNet (Krizhevsky et al., 2012) and ResNet-18 (He et al., 2016), both of which are popular in classifying image data (Alom et al., 2018). We mainly show the experimental results of predicting *wavy hair* using AlexNet. *More results and training details are in supplementary materials.*

**Metrics.** For fairness evaluation, we take two group fairness metrics DP and EO as we introduced in the section 3.2 and define $-$DP and $-$EO as fairness scores since smaller DP and EO mean better fairness. We use the average precision (AP) for classification accuracy evaluation. A desired fairness method should achieve smaller DP or EO but higher AP (*i.e.*, the top left corner in Fig. 3).

**Baselines.** Following the common setups in Chuang & Mroueh (2021), we compare our method with several baselines: ❶ Standard training based on empirical risk minimization (ERM) principle (*i.e.*, w.o.FairReg). DNNs are trained only with the cross entropy loss. ❷ Oversample (*i.e.*,

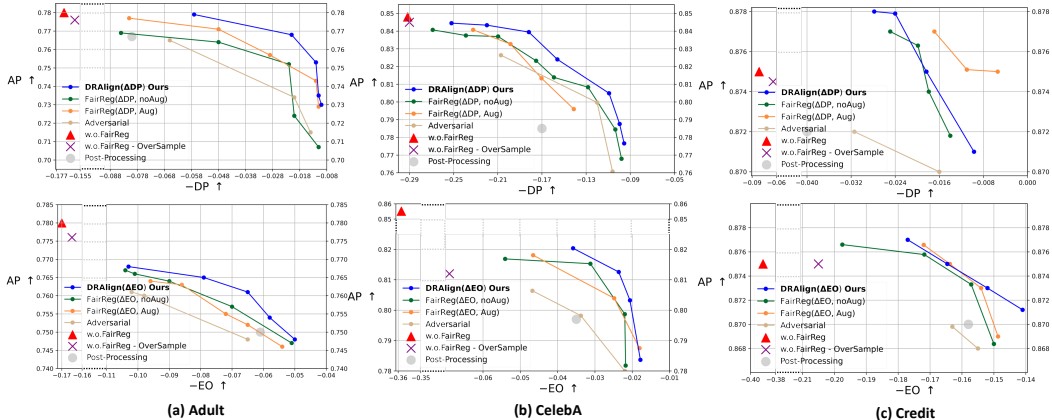

Figure 3: Comparing different methods on AP vs. (-DP/-EO). According to the common setups, we evaluate $\Delta DP$-based and $\Delta EO$-based methods via DP and EO, respectively. We train networks with the compared methods for 10 times and the averaging results are reported.

w.o.FairReg-Oversample) (Wang et al., 2020). This method samples from the subgroup with rare examples more often, making a balanced sampling in each epoch. ❸ FairReg($\Delta$DP or $\Delta$EO, noAug) (Madras et al., 2018). This method is to directly regularize the fairness metrics, *i.e.*, $\Delta$DP or $\Delta$EO. ❹ FairReg($\Delta$DP or $\Delta$EO, Aug) (*i.e.*, FairMixup) (Chuang & Mroueh, 2021). This method regularizes the models on paths of interpolated samples between subgroups to achieve fairness. ❺ Adversarial (Zhang et al., 2018). This method minimizes the adversary's ability to predict sensitive attributes. ❻ Post-processing (Bellamy et al., 2018). This method modifies the predictions of an accurate model with a fairness objective.

## 6.2 FAIRNESS IMPROVEMENT PERFORMANCE

As shown in Fig. 3, we have following observations: ❶ With the Adult and CelebA datasets, our method (*i.e.*, DRAlign) achieves higher fairness (*i.e.*, higher -DP or -EO scores) than all baseline methods when they have similar accuracy. In particular, on the Adult dataset, DRAlign has relative $41.6\%$ DP improvement over the second best method (*i.e.*, FairReg($\Delta$DP, Aug)) when both get around 0.770 AP. Overall, our method can enhance the fairness significantly with much less accuracy sacrifice. ❷ Data augmentation method does not always improve DNN's fairness. For example, on the dataset Adult, FairReg($*$, noAug) presents slightly higher fairness score (*i.e.*, higher -DP or -EO) than FairReg($*$, Aug). A potential reason is that the augmented data becomes less realistic due to the rich information in the image modality, which leads to less effective learning. ❸ Although oversampling could improve fairness to some extent, it is less effective than the fairness regularization-based methods (*i.e.*, FairReg($*$, noAug)). For example, on the CelebA dataset, w.o.FairReg-Oversample only obtains -0.069 -EO score with the 0.812 AP score, while FairReg($*$, noAug) achieves the -0.054 -EO score with 0.817 AP score. The networks trained by FairReg($*$, noAug) are not only fairer but also of higher accuracy. On the tabular dataset, w.o.FairReg-Oversample outperforms the w.o.FairReg by a small margin. ❹ On the Credit dataset, FairReg($\Delta DP$, Aug) achieves better results than DRAlign under the DP metric although our method still outperforms the regularization-based one. A potential reason is that the data size of the Credit is small (*i.e.*, 500 training samples) and the data augmentation can present obvious advantages by enriching the training data significantly. The data augmentation and our decision rationale alignment are two independent ways to enhance fairness. Intuitively, we can combine the two solutions straightforwardly. We do further experiments and find that our DRAlign could still improve FairReg($\Delta DP$, Aug). More details are put in supplementary materials.

## 6.3 DISCUSSION AND ANALYSIS

**Connection with over-parameterization.** To better understand the cause of the decision rationale misalignment, we further investigate the connection between decision rationale misalignment and model over-parameterization. We conduct an empirical study on the Adult dataset using 3-layer MLP networks based on FairReg($\Delta$DP, noAug). Specifically, we explore 4 MLP architectures, where the hidden sizes are set as 10, 20, 50, and 200, respectively. The corresponding parameter sizes of the 4 networks are 1331, 2861, 8651, and 64601. For each architecture, we draw a plot w.r.t. different $\lambda$ for

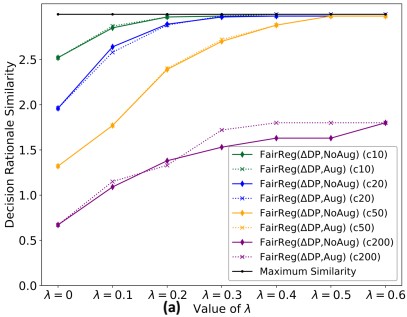 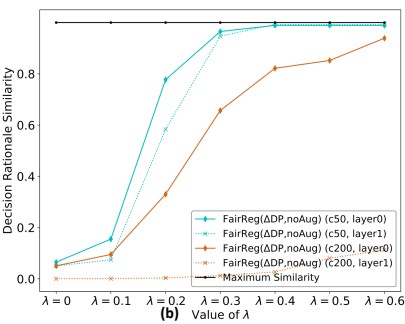

Figure 4: (a) : correlation between $\lambda$ and decision rationale similarity score. (b) layer-wise analysis for correlation between $\lambda$ and decision rationale similarity score.

FairReg($\Delta$DP, noAug) to show the decision rationale similarity score (*i.e.*, $\sum_{l=0}^{L} \cos(\vec{c}_l^{a=0}, \vec{c}_l^{a=1})$ in Sec. 5.2). We denote the four trained models as FairReg($\Delta$DP,noAug) (c10), FairReg(noAug) (c20), FairReg($\Delta$DP,noAug) (c50), and FairReg($\Delta$DP,noAug) (c200), respectively, according to their hidden sizes. *More results under $\Delta EO$ metric are put in Supplementary Materials.* With Fig. 4 (a), we have the following observations: ❶ The decision rationale similarity consistently ascends when $\lambda$ increases. When $\lambda$ becomes 0.5, decision rationale similarities of FairReg($\Delta$DP,noAug) (c10), FairReg($\Delta$DP,noAug) (c20)and FairReg($\Delta$DP,noAug) (c50) reach the same maximum score (*i.e.*, 3.0 for any 3-layer MLP network). We conclude that larger $\lambda$ (stricter fairness constraint) results in a higher decision rationale similarity. ❷ The misalignment of decision rationale is more likely to occur in the over-parameterized networks. For the largest network FairReg($\Delta DP$,noAug) (c200), even when the $\lambda$ is set as 0.6 for a strict fairness constraint, the decision rationale similarity score only reaches 1.8 which is much smaller than the values on other variants and infers that the decision rationale is still far from being aligned.

Furthermore, we also report the results of augmentation-based method, *i.e.*, FairReg($\Delta$DP,Aug). We find that data augmentation can generally mitigate the misalignment but still fails to completely align the decision rationale (See the plot of FairReg($\Delta DP$,noAug) (c200)). Our method DRAlign is able to achieve the maximum similarity on all $\lambda$ settings even on the architecture with hidden size 200. This enlightens us that common methods such as data augmentation that aims to address over-parameterization can not completely solve the misalignment, while our gradient-guided parity alignment method can directly improve the alignment.

**Layer-wise decision rationale alignment analysis.** We further conduct a layer-wise analysis to understand which layer owns better decision rationale alignment. We calculate the decision rationale similarity for the 1st and 2nd layer (*i.e.*, $\cos(\vec{c}_{l=0}^{a=0}, \vec{c}_{l=0}^{a=1})$ and $\cos(\vec{c}_{l=1}^{a=0}, \vec{c}_{l=1}^{a=1})$). From Fig. 4 (b), we see that: for both layers, the layer-wise similarity score ascends when $\lambda$ increases. This is consistent with the observation that stricter fairness constraint results in a higher decision rationale similarity. As we compare the 1st and 2nd layers, we can observe that the similarity score of the first layer is generally higher. Moreover, we can see that for smaller models (*i.e.*, models with hidden size 50), the similarity gap between the first layer and the second layer is relatively trivial. However, for models with hidden size 200, the similarity score of the second layer is rather low (*i.e.*, the score is 0.113 even when the $\lambda$ is 0.6). We conclude that the misalignment of the deeper layer is severer.

# 7 CONCLUSIONS AND FUTURE WORK

In this work, we have studied the fairness issue of deep models from the perspective of decision rationale and defined the *neuron parity score* to characterize the decision rationale shifting across subgroups. We observed that such a decision rationale-aware characterization has a high correlation to the fairness of deep models, which means that a fairer network should have aligned decision rationales across subgroups. To this end, we formulated fairness as the decision rationale alignment (DRAlign) and proposed the *gradient-guided parity alignment* to implement the new task. The results on three public datasets demonstrate the effectiveness and advantages of our methods and show that DRAlign is able to achieve much higher fairness with less accuracy sacrifice than all existing methods. **Although promising, our method also presents some drawbacks: (1) it requires the computation of second-order derivatives; and (2) the gradient-guided parity alignment method is limited to the layer-wise DNN architecture. In the future, we are interested in solving these limitations.**

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
