# OpenReview forum: "FAIRER: Fairness as Decision Rationale Alignment"
_ICLR.cc/2023/Conference — Submitted to ICLR 2023_

### Official Review · Reviewer_X7LC · 2022-10-16

**Confidence:** 5
**Correctness:** 3
**Technical Novelty And Significance:** 2
**Empirical Novelty And Significance:** 3
**Recommendation:** 6

**Clarity, Quality, Novelty And Reproducibility:**

The writeup of the paper is very clear. There are some missing hyperparameter details, the completion of which can aid in assessing the reproducibility of the work.

**Strength And Weaknesses:**

1. The proposed method can be easily integrated in the training procedure for an arbitrary network, making it highly generalizable.
2. Experiments have been performed across multiple values of $\lambda$ and clearly details the pareto frontier.
3. The subsection on “Layer-wise Decision Rationale Alignment” provides interesting characterization against the depth of the model.
4. An important citation to  (Zheng et al 2021; NeuronFair: Interpretable White-Box Fairness Testing through Biased Neuron Identification) is missing. This work has some similarities to that of Zheng et al, in that both focus on proposing a function over the network neurons.
5. The computation of the second order derivatives is a significant overhead. It would be interesting to see some results comparing the runtime of the proposed method against the baselines.
6. While computing eq 7, do the authors turn off the dropout layer in the model? Not turning it off could have detrimental results on the performance.
7. There is no study for the variation of results against $\beta$, which is the hyperparameter of the proposed misalignment objective.
8. The complete hyperparameter details such as the optimizer used, batch size, weight decay etc are missing.
9. Using cosine similarity in eq 9 is a reasonable strategy. Is it chosen for simplicity? Do the authors have some intuition on how the cosine will compare against other metrics, a simple one being the metric induced by the L2-norm?
10. The final objective in eq 9 utilizes a fairness regularizer as well. It will be more useful to observe the results solely based on the cross-entropy loss along with the proposed alignment regularizer to ensure that the claims hold.


**Summary Of The Paper:**

The authors propose a new mechanism, called *gradient-guided parity alignment* to ensure fairness in neural models. This is done by studying the fairness problem from the perspective of decision rationale, which has been one of the mainstream areas of focus in interpretability literature. The authors implement their proposed mechanism along with the standard fairness regularizer which is the mainstream strategy in the in-processing group fairness literature. More concretely, first the loss change before and after removing a specific parameter $w_k$ of the neural model if computed in eq 4. This is then used to define the *neuron parity score* in eq 5. Intuitively, this characterizes that the change in the loss based on a given $w_k$ should be the same across the subgroups. The parity score is computed for all the parameters in the model. Furthermore, since directly computing this for all samples and neurons is expensive, the authors utilize a sampling strategy. Eq 7 is relaxed to obtain the gradient weighted estimation of the change in loss based on the first order Taylor expansion. Lastly, these parity scores are normalized layerwise and used in the final loss objective, with the underlying parity computed based on cosine similarity. Extensive experiments have been performed against popular fairness methods along with interesting analyses.


**Summary Of The Review:**

Based on the questions, comments and concerns raised in the weakness section, I lean towards weak rejection of the work. I am willing to reconsider my score if the authors can answer the questions raised above.

---

> ### Author Response · Authors · 2022-11-19
> **Response to Reviewer X7LC (2/2)**
>
> 3. **While computing eq 7, do the authors turn off the dropout layer in the model? Not turning it off could have detrimental results on the performance.**
>
> When evaluating the neuron parity score, we turn dropout off and fix the parameters in BN layer. During the training process, we didn’t turn dropout off, which didn’t cause detrimental results in our experiments. Our setting follows the implementation in [0]. We think the optimization is based on a group of data, which statistically calculates the average prediction and gradients and won’t cause detrimental results.
>
> 4. **There is no study for the variation of results against β, which is the hyperparameter of the proposed misalignment objective.**
>
> Thanks for the question. In our paper, we did a rough search for the hyper-parameter β. Taking CelebA dataset as an example, we mainly search β value in the range {0.001, 0.01, 0.1}. When β is set as 0.001, the training process is close to that of FairReg, which means that our decision rationale alignment item is ignored in the training because β is too small. When β is 0.1, the training process will optimize the decision rationale alignment first and cause detrimental influence on the optimization of other loss items. We finally choose 0.01 as the β value. We have added the above discussion in the supplementary materials A.1.
>
> 5. **The complete hyperparameter details such as the optimizer used, batch size, weight decay etc are missing.**
>
> For the Adult dataset, we use Adam as the learning optimizer and the batch size is set as 1000 for the DP metric and 2000 for the EO metric. For the CelebA dataset, we use Adam as the learning optimizer and the batch size is set as 64 for the DP metric and 128 for the EO metric. These settings follow [0]. For the Credit dataset, We use Adam as the learning optimizer and the batch size is set as 400 for the DP metric and 500 for the EO metric.
> We add these details to our supplementary materials A.1.
>
> 6. **Using cosine similarity in eq 9 is a reasonable strategy. Is it chosen for simplicity? Do the authors have some intuition on how the cosine will compare against other metrics, a simple one being the metric induced by the L2-norm?**
>
> We do normalization to the $c_l$ vector and the L2-norm of the normalized vectors equals the cosine similarity between $c_l$ vectors.  Moreover, compared with L2-norm, cosine similarity owns two advantages. (1) When evaluating the distance between two vectors, cosine similarity provides definite bounds. The upper bound of cosine similarity is 1 and the lower bound is 0, which is easier for us to compare the decision rationale alignment degree across different subgroups or models (as shown in Figure 4). (2) When optimizing the model with the L2-norm loss item, the optimization might suppress the $c_l$ value as introduced in [1] (the optimization of L2-norm suppresses the activation). The suppression might interfere the normal training process. Here we use cosine similarity calculation.
>
> 7. **The final objective in eq 9 utilizes a fairness regularizer as well. It will be more useful to observe the results solely based on the cross-entropy loss along with the proposed alignment regularizer to ensure that the claims hold.**
>
> Thanks for the comments. We find that the alignment itself could still slightly improve fairness when fairness regularization is removed. Specifically, we remove the $L_\text{fair}$  term in Eq.(6) and retain the classification loss and the decision rationale alignment loss. We denote this version as $\text{w.o.FairReg-DRAlign}$. Compared with the model only trained with the classification loss (i.e., $\text{w.o.FairReg}$ ), $\text{w.o.FairReg-DRAlign}$ increase the experimental results (AP, -DP) from (0.776, -0.16 ) to (0.781, -0.14). The results are consistent with our observation that our decision rationale alignment method could further improve fairness and demonstrate that decision rationale alignment is actually a favorable supplement for existing fairness regularization terms. We have added this discussion to Figure 2 and Supplementary Materials A.9.
>
> &nbsp;
>
> &nbsp;
>
> [0] Ching-Yao Chuang and Youssef Mroueh. Fair mixup: Fairness via interpolation. In International Conference on Learning Representations, 2021.
>
> [1] Harini Kannan, Alexey Kurakin, Ian Goodfellow. Adversarial Logit Pairing. arXiv preprint 2018 https://arxiv.org/abs/1803.06373

---

> ### Author Response · Authors · 2022-11-19
> **Response to Reviewer X7LC (1/2)**
>
> Thank you for the constructive comments and advice. Our detailed responses are shown below:
>
> 1. **An important citation to (Zheng et al 2021; NeuronFair: Interpretable White-Box Fairness Testing through Biased Neuron Identification) is missing. This work has some similarities to that of Zheng et al, in that both focus on proposing a function over the network neurons.**
>
> Thanks for providing the related work.
> NeuronFair is to analyze neurons sensitive to individual discrimination and generate testing cases according to sensitive neuron behaviors.
> Different from the NeuronFair work, our work focuses on group fairness rather than individual discrimination. The function we define as biased neurons is based on neuron influence rather than activation values. Moreover, our work provides a DRAlign method to repair the model not only testing. Even so, The NeuronFair paper well enlightens us. We cite this paper in our updated version.
>
>
> 2. **The computation of the second-order derivatives is a significant overhead. It would be interesting to see some results comparing the runtime of the proposed method against the baselines.**
>
> We further add the estimation of the computational cost into our paper. Please be noted that the FairReg(Aug) method also requires the calculation of a second-order derivative. Moreover, as a method based on data augmentation, the FairReg(Aug) method requires more time to converge. Here we show the time cost of using CelebA dataset and AlexNet under the DP metric. More results could be found in the supplementary materials. We have added a discussion in the supplementary materials A.5.
>
> |       | w.o. FairReg | w.o. FairReg-Oversample | FairReg(\*, noAug)| FairReg(\*, Aug)| DRAlign |
> |:----: | :----:| :----:| :----:| :----:| :----:|
> |Training Time  | 611s | 725s | 811s | 1995s | 1397s |

---

### Official Review · Reviewer_rJaT · 2022-10-22

**Confidence:** 4
**Correctness:** 3
**Technical Novelty And Significance:** 3
**Empirical Novelty And Significance:** 2
**Recommendation:** 5

**Clarity, Quality, Novelty And Reproducibility:**

**Novelty and quality. **

The work is novel, in my opinion. Imposing the Neuron level to achieve parity is interesting and this work provides some additional insight and an algorithmic solution for learning tasks. Overparametrization results, while incomplete, are novel and interesting. This work is well-written and easy to follow (see the following for the typos and suggestions).

** Reproducibility. **

What is the value $\beta$ in Eq. (9) in the experiments? Are the results sensitive to the choice of $\beta$?
$\lambda$ is discussed in detail but not $\beta$, am I missing something?

**Clarity.**

Page 1.

Sec 1. What does social fairness in line one of the introduction mean?

Page 3.

Sec 3.1. "A fair DNN (i.e., F(·)) is desired to obtain a similar accuracy in the two subgroups." This is only a definition of fairness, that is suboptimal and leads to model underperformance.

Page 4.

Sec 3.3.
Does the background color gradient in Figure 2 mean anything?

"(i.e., smaller DP)" do the authors mean larger DP or smaller absolute value of DP?

❷ --> ❸

"The data augmentation-based method (i.e., FairReg(∆DP, Aug)) can alleviate such an issue to some extent and achieves higher fairness than FairReg(∆DP, noAug) under similar accuracy." This is not consistent with Figure 2.

Page 5.

Section 4.1

$$\mathcal{J}$$ is not defined where it first appears.

Page 6.

Section 5.2

"Then, we can update Eq. (5) by minimizing the" --> do you mean Eq. (6)?





**Details Of Ethics Concerns:**

Fairness is not put into context. It is not argued why degrading accuracy is desired when there is no decision-making component.

**Strength And Weaknesses:**

Strength:
* This work is easy to follow and clear.
* AI Fairness is an important, emerging topic.
* The insight from network size and fairness is valuable.
* Imposing fairness regularization at the neuron level that is computationally scalable is interesting.

Weakness:
* Prediction is confused with decision-making (see below).
* This work does not provide an argument why parity in the accuracy of prediction, specifically in their experiments, is desired at a cost of lowering the accuracy (see below).
* The error bars are not provided, so it is hard to evaluate the statistical significance of the results (e.g., Figure 3). The results seem marginal and statistically insignificant.
* This work does not discuss the limitations of this work.
* Computational costs are not discussed nor how it is compared with the other methods.


**More details.**

1. Fairness, prediction, and decision-making.

This work is concerned with the parity of accuracy across protected groups in a prediction task. As an example, they ask how well a model can predict the income level (binary prediction >50k or below) of male and female participants. There is no decision-making downstream component here though. So I do not understand why parity is the accuracy of prediction is considered a fairness criterion. Potentially there is more intrinsic stochasticity in the income level of the female labor force participants and less stochasticity in male participants. This inherent feature reduces the accuracy of a classifier for female participants. By imposing an accuracy parity constraint we essentially degrade the quality of our prediction for male participants while keeping the quality of prediction for the female participant the same, hence degrading total accuracy. Why is this a desired feature? If there is no decision-making component why the authors are after a degraded classifier?

In the section *Connection with human society*, prediction task that is the focus of this work is confused with the decision-making task. While a classifier can be used as an ingredient in a decision-making task the connection to decision-making and what type of problem is solved here is vague. Suppose I want to predict which galaxy is star-forming and which galaxy is non-star-forming and I have a prediction disparity between spiral galaxies and elliptical galaxies. Is this fair or unfair? But now I can ask how to allocate telescope resources for a set of science applications; then we can actually discuss the fairness in the allocation of resources even though we are talking about galaxies. Now, let's go back to the income prediction example. If a classifier has a poor performance for the female participant, is that classifier unfair? a classifier cannot be intrinsically fair or unfair without specifying the downstream decision-making task, especially if we achieved the optimal Bayes classifier.

2. Lack of confidence intervals.

The results are marginal and improve the previous methods only by a small margin with a significant additional computational cost (even though the proposed algorithm is scalable). This work should provide error bars for the results so the reader can understand whether the results are significant and whether the additional computational cost is worth the improvement. Without confidence intervals, it is impossible to discuss the significance of the results.

3. Connection with over-parameterization.

This section actually provides an interesting insight. However, the comparison is incomplete. Still, we do not know how the accuracy degrades across different models. Does the accuracy also converge across over parameterized models or does the accuracy differs? Hence, a designer does not know whether they should shoot for overparameterization and larger regularization or lower parameterization and smaller regularization.


**Summary Of The Paper:**

This work proposed a novel method of training a fair classifier. In this work, the authors are concerned about the parity of accuracy across protected groups and propose a new parity score that is computationally manageable at a level of individual neurons. This work illustrates the advantages of this method in several empirical settings.

**Summary Of The Review:**

While this work has potential and is interesting, there are areas that could be improved. Specifically,
(1) what kind of decision problems is this work concerned about and why the choice of fairness metrics are desired? (2) what is the statistical significance of the results (confidence intervals)? (3) what are the computational costs and how it is compared with the prior methods? how the accuracy changed with $\lambda$ and the parameter size.

---

> ### Author Response · Authors · 2022-11-19
> **Response to Reviewer rJaT (2/2)**
>
>
>
> 5. **This section actually provides an interesting insight. However, the comparison is incomplete. Still, we do not know how the accuracy degrades across different models. Does the accuracy also converge across over parameterized models or does the accuracy differs? Hence, a designer does not know whether they should shoot for overparameterization and larger regularization or lower parameterization and smaller regularization.**
>
> Thanks for the interesting question. We show the best ap performance of different models via the validation set. We can see that larger models are pruned to have higher APs. In our work,  we provide an analysis method to analyze the decision rationale consistency and propose the gradient-guided parity alignment method to solve the misalignment. Models with smaller parameterization might suffer from lower accuracies. If the designer shoots for over-parameterization, we can analyze whether the misalignment exists in the model and repair the misalignment to improve fairness.  We add the experimental results to our supplementary materials A.6.
>
> |  $\lambda$     | $\lambda=0$ | $\lambda=0.1$ | $\lambda=0.2$ | $\lambda=0.3$ | $\lambda=0.4$ | $\lambda=0.5$ | $\lambda=0.6$ | $\lambda=0.7$ |
> |:------: | :------:| :------:| :------:| :------:| :------:| :------:| :------:| :------:|
> |$c_{10}$ | 0.781 | 0.780 | 0.776 | 0.768 | 0.758 | 0.745 | 0.731 | 0.729 |
> |$c_{20}$ | 0.782 | 0.780 | 0.777 | 0.768 | 0.757 | 0.743 | 0.734 | 0.728 |
> |$c_{50}$ | 0.783 | 0.781 | 0.776 | 0.769 | 0.758 | 0.741 | 0.737 | 0.730 |
> |$c_{200}$ | 0.784 | 0.781 | 0.777 | 0.769 | 0.760 | 0.743 | 0.744 | 0.738 |
>
>
> 6. **What is the value β in Eq. (9) in the experiments? Are the results sensitive to the choice of β? λ is discussed in detail but not β, am I missing something?**
>
> Thanks for the question. In our paper, we did a rough search for the hyper-parameter β. Taking CelebA dataset as an example, we mainly search β value in the range {0.001, 0.01, 0.1}. When β is set as 0.001, the training process is close to that of FairReg, which means that our decision rationale alignment item is ignored in the training because β is too small. When β is 0.1, the training process will optimize the decision rationale alignment first and cause detrimental influence on the optimization of other loss items. We finally choose 0.01 as the β value. We have added the above discussion in the supplementary materials A.1.
>
> 7. **Other detailed questions**
>
> * **What does social fairness in line one of the introduction mean?**
>
> We explain more about the word “social fairness” in the introduction. In our social life, there are policies requiring people of different genders/races/ages to own the same opportunities/possibilities in case of being accused of discrimination. For example, for an intelligent task (e.g., salary prediction), a trained DNN easily presents distinct accuracy values in different subgroups (e.g., male and female).
>
> * **Sec 3.1. "A fair DNN (i.e., F(·)) is desired to obtain a similar accuracy in the two subgroups." This is only a definition of fairness, that is suboptimal and leads to model underperformance.**
>
> Our paper follows the setting in [0][1][2]. We mainly discuss fairness under such a definition.
>
> * **Sec 3.3. Does the background color gradient in Figure 2 mean anything?**
>
> The background color is for a better appearance to highlight the best performance regions, i.e., right-top corner.
>
> * **"(i.e., smaller DP)" do the authors mean larger DP or smaller absolute value of DP?**
>
> It means “smaller DP score” (absolute value). That is a larger -DP score. We have revised the descriptions.
>
> * **"The data augmentation-based method (i.e., FairReg(∆DP, Aug)) can alleviate such an issue to some extent and achieves higher fairness than FairReg(∆DP, noAug) under similar accuracy." This is not consistent with Figure 2.**
>
> Thanks for your careful reading. In Figure2, we got the line color wrong. We revise this part in our new version and we do further proofreading.
>
> * **Section 4.1 J is not defined where it first appears. "Then, we can update Eq. (5) by minimizing the" --> do you mean Eq. (6)?**
>
> Thanks for your reminder. We add the description of J and revise “Eq. (5)” to “Eq.(6)”, and we do further proofreading.
>
> &nbsp;
>
> &nbsp;
>
>
> [0] Ching-Yao Chuang and Youssef Mroueh. Fair mixup: Fairness via interpolation. In International Conference on Learning Representations, 2021.
>
> [1] Mengnan Du, Subhabrata Mukherjee, Guanchu Wang, Ruixiang Tang, Ahmed Hassan Awadallah, and Xia Hu. Fairness via representation neutralization. In NeurIPS, 2021.
>
> [2] Wang, G., Du, M., Liu, N., Zou, N., and Hu, X., “Mitigating Algorithmic Bias with Limited Annotations”, TPAMI, 2022.
>
> [3] Kleinberg, Jon & Mullainathan, Sendhil & Raghavan, Manish. (2016). Inherent Trade-Offs in the Fair Determination of Risk Scores.

---

> ### Author Response · Authors · 2022-11-19
> **Response to Reviewer rJaT (1/2)**
>
> Thank you for the constructive comments and advice. Our detailed responses are shown below:
>
> 1. **Prediction is confused with decision-making (see below). This work does not provide an argument why parity in the accuracy of prediction, specifically in their experiments, is desired at a cost of lowering the accuracy (see below).**
>
> Thanks for the comments. First,  the fairness of deep models is critical in real-world applications and  has been carefully studied in many previous works [0][1][2]. In our social life, there are policies requiring people of different genders/races/ages to own the same opportunities/possibilities in case of being accused of discrimination. For example, for an intelligent task (e.g., salary prediction), a trained DNN easily presents distinct accuracy values in different subgroups (e.g., male and female). Also as said in [1], training a model on COMPAS, is likely to associate African-American offenders with higher risk scores compared to Caucasians while having a similar profile.  Second, state-of-the-art fairness works [1][2] add fairness regularization terms to the loss function, which can enhance fairness but decrease the accuracy significantly.  As introduced in [3], there is an inherent trade-off between fairness and accuracy.  In this work, we study fairness from the perspective of decision rationale that characterizes the decision-making process and we try to reduce the cost of lowering the accuracy by aligning the decision rationales of different subgroups (e.g., female and male).  Note that, the prediction of a network and the decision-making process are different. In general, the prediction represents the final outputs of a network such as the category predicted by a classification network. The decision-making process or the decision rationale is to represent how the output prediction is made. In this work, inspired by recent works on DNN understanding, we propose the neuron parity score to measure the importance of intermediate neurons to the fairness of the whole network.
>
> 2. **The error bars are not provided, so it is hard to evaluate the statistical significance of the results (e.g., Figure 3). The results seem marginal and statistically insignificant.**
>
> We follow the exhibition method in [0][1][2]. We here report the average result over 10 times. In the updated version, we additionally provide the error bar in the supplementary materials A.11 and we also provide the error bar to our tabular statistics as shown in Table 1.
>
> 3. **This work does not discuss the limitations of this work.**
>
> The reviewer seems to ignore our discussion in the conclusion section. We highlight the limitations in the conclusion section.
>
> 4. **Computational costs are not discussed nor how it is compared with the other methods.**
>
> We further add the estimation of the computational cost into our paper. Please be noted that the FairReg(Aug) method also requires the calculation of a second-order derivative. Moreover, as a method based on data augmentation, the FairReg(Aug) method requires more time to converge. Here we show the time cost of using CelebA dataset and AlexNet under the DP metric. More results could be found in the supplementary materials. We have added a discussion in the supplementary materials A.5.
>
> |       | w.o. FairReg | w.o. FairReg-Oversample | FairReg(\*, noAug)| FairReg(\*, Aug)| DRAlign |
> |:----: | :----:| :----:| :----:| :----:| :----:|
> |Training Time  | 611s | 725s | 811s | 1995s | 1397s |

---

> ### Author Response · Authors · 2022-11-23
> **Message from Authors**
>
> Since we're still in the second stage of discussion, we'd like to know what the reviewer thinks of our revision. Has our response helped the reviewer address the reviewer's concerns and reconsider the value of our work? We still have time to discuss if there are any new issues, and we'll be pleased to address them.
>
> We are looking forward to hearing from the reviewer.
>
> Thanks.

---

### Official Review · Reviewer_HMGh · 2022-10-23

**Confidence:** 3
**Correctness:** 3
**Technical Novelty And Significance:** 3
**Empirical Novelty And Significance:** 3
**Recommendation:** 5

**Clarity, Quality, Novelty And Reproducibility:**

I am generally satisfied with the clarity and quality of writing. The proposed method seems to be novel and reproducible (the code has been provided).

**Strength And Weaknesses:**

__Strength__

- The proposed method is versatile, in the sense that it can be provided to several different notions of fairness.
- The method shows an impressive performance over the baselines in most empirical setups considered in the paper.
- Empirical validation has been performed in terms of the full tradeoff curve, instead of showing only on data point.

__Weakness__

- The paper argues that it will give an analysis of the existing fairness-via-regularization technique via decision rationale analysis, but I do not see it. I can only see the experimental results showing that the existing method performs worse. Am I missing anything?

- The "limitations" pointed out in Section 3.3 is not very clear to me. The paper says that existing methods are limited, as they could "hardly generate enough fair networks under similar accuracy." But the "enough"-ness is in the eye of the beholder; why is 0.058 DP enough, and 0.078 not enough? The logic sounds like, 'existing methods are limited, as it does not perform as well as the method that we are going to propose, which motivates us to develop a new method.' This does not really point out any fundamental, structural limitations of the existing methods.

- The intuition behind using the neuron parity score for fairness is unclear. Authors seem to motivate the technique from the fact that "how the fairness regularization term affect the final network parameter is not well understood," but I am not sure how the existing method provides a strictly better interpretation, or why such better parameter-wise interpretability should help improving the ultimate performance.

- The range of baselines considered is quite limited. There is a wealth of methods for fair classification, including the ones based on adversarial debiasing, optimal transport, etc. Is there a reason why authors confined the baselines to only a few?

- Eq (5) is quite atypical, in the sense that it is squaring the already-squared quantities. Although this is not really used in the main algorithm, I am curious about the reason why such nested squaring is a desirable thing to do. Could authors give a little more explanation?

**Summary Of The Paper:**

This paper proposes a new fairness regularization technique based on an interpretation method called neuron parity score. In a nutshell, the proposed method penalizes the gap among the _subgroup loss increments_ from zero-ing out a specific parameter, in addition to the model-level fairness loss. Empirically, the proposed method achieves a better fairness-performance frontier than FairMixup and LAFTR.

**Summary Of The Review:**

Although the method seems to provide decent performance gain over the baselines, I think there is a significant room for the manuscript to improve in terms of (1) the diversity of the baseline methods, and (2) clarity of the motivation behind the algorithm.

---

> ### Author Response · Authors · 2022-11-19
> **Response to Reviewer HMGh (2/2)**
>
> 3. **The intuition behind using the neuron parity score for fairness is unclear. Authors seem to motivate the technique from the fact that "how the fairness regularization term affect the final network parameter is not well understood," but I am not sure how the existing method provides a strictly better interpretation, or why such better parameter-wise interpretability should help improving the ultimate performance.**
>
> Sorry for the unclear statement. Our initial draft was to introduce the FairReg method first and show that this method ignores the decision rationale which restricts the fairness performance. We tried to clarify the limitations in the front section before introducing the neuron parity score and decision rationale analysis to make our purpose clearer. We realize now that this makes the writing logic confusing. We appreciate your careful reading and we reorganize the writing in the last paragraphs of sections 3.3 and 4.2.
>
> 4. **The range of baselines considered is quite limited. There is a wealth of methods for fair classification, including the ones based on adversarial debiasing, optimal transport, etc. Is there a reason why authors confined the baselines to only a few?**
>
> Thanks for your advice. In our submission, we have tried our best to compare with all fairness works. To address the concern,  we add two more baselines in Figure 3, i.e., debias-related work [3] and post-processing-based work [4]. In the previous work [0][1][2] we referred to, we observed that the included baselines, .e.g., FairReg method, are consistently better than the newly added baseline methods. That is why we ignored them in our initial submission.
> We report new experimental results in the updated version. The experimental results show that our method still outperforms the new baseline methods.
>
> 5. **Eq (5) is quite atypical, in the sense that it is squaring the already-squared quantities. Although this is not really used in the main algorithm, I am curious about the reason why such nested squaring is a desirable thing to do. Could authors give a little more explanation?**
>
> Thanks for your question. Here we choose the squared function as a distance evaluation method. Such distance evaluation methods should not influence the evaluation results.
>
> &nbsp;
>
> &nbsp;
>
> [0] Ching-Yao Chuang and Youssef Mroueh. Fair mixup: Fairness via interpolation. In International Conference on Learning Representations, 2021.
>
> [1] Mengnan Du, Subhabrata Mukherjee, Guanchu Wang, Ruixiang Tang, Ahmed Hassan Awadallah, and Xia Hu. Fairness via representation neutralization. In NeurIPS, 2021.
>
> [2] Wang, G., Du, M., Liu, N., Zou, N., and Hu, X., “Mitigating Algorithmic Bias with Limited Annotations”, TPAMI, 2022.
>
> [3] Z. Wang, K. Qinami, I. Karakozis, K. Genova, P. Nair, K. Hata, and O. Russakovsky. Towards fairness in visual recognition: Effective strategies for bias mitigation. In 2020 IEEE/CVF Conference on Computer Vision and Pattern Recognition (CVPR), pp. 8916–8925, Los Alamitos, CA, USA, jun 2020. IEEE Computer Society.
>
> [4] Rachel K. E. Bellamy, Kuntal Dey, Michael Hind, Samuel C. Hoffman, Stephanie Houde, Kalapriya Kannan, Pranay Lohia, Jacquelyn Martino, Sameep Mehta, Aleksandra Mojsilovic, Seema Nagar, Karthikeyan Natesan Ramamurthy, John Richards, Diptikalyan Saha, Prasanna Sattigeri, Moninder Singh, Kush R. Varshney, and Yunfeng Zhang. AI Fairness 360: An extensible toolkit for detecting, understanding, and mitigating unwanted algorithmic bias, October 2018.

---

> ### Author Response · Authors · 2022-11-19
> **Response to Reviewer HMGh (1/2)**
>
> Thank you for the constructive comments and advice. Our detailed responses are shown below:
>
> 1. **The paper argues that it will give an analysis of the existing fairness-via-regularization technique via decision rationale analysis, but I do not see it. I can only see the experimental results showing that the existing method performs worse. Am I missing anything?**
>
> Thanks for pointing out the confusion. The reviewer seems to neglect the contents in Section 4.2. In section 3, we show that existing works adding fairness regularization terms to the loss function can increase the fairness but sacrifice the accuracy significantly (See Fig. 2). Existing works only provide fairness metric results but neglect the decision-making process. In section 4.1, we provide an analysis method by extending the decision rationale-aware explainable methods. Specifically, instead of using the final metrics, we define the neuron parity score for each parameter of a network that measures whether the parameter is fair, that is, whether it has consistent responses to different subgroups. With this analysis method, we conduct a study in section 4.2 and reveal
> the strong connection between neuron parity score and fairness.
> (See Tab. 1): the existing fairness regularization-based methods can encourage the consistency  between decision rationales of the network on different subgroups to some extent and get smaller neuron parity scores than the standard trained network. This inspires our method in Sec. 5 that conducts alignments of the decision rationales of different subgroups explicitly. Although intuitively, the consistent decision process of different groups could improve the fairness performance, to explore the concrete relationship between the decision-making process and fairness, we propose the decision rationale-aware fairness analysis in section 4. To avoid confusion, we rewrite the last paragraphs of sections 3.3 and 4.2.
>
> 2. **The "limitations" pointed out in Section 3.3 is not very clear to me. The paper says that existing methods are limited, as they could "hardly generate enough fair networks under similar accuracy." But the "enough"-ness is in the eye of the beholder; why is 0.058 DP enough, and 0.078 not enough? The logic sounds like, 'existing methods are limited, as it does not perform as well as the method that we are going to propose, which motivates us to develop a new method.' This does not really point out any fundamental, structural limitations of the existing methods.**
>
> Thanks for the comments. We do not aim to provide insight analysis in section 3.3 but to show that existing fairness regularization terms compared with the standard trained model can enhance the fairness but harm the accuracy. Existing works only provide fairness metric results but neglect the decision-making process like Fig. 2. In section 4.1, we provide an analysis method by extending the decision rationale-aware explainable methods. With this analysis method, we conduct a study in section 4.2 and reveal the strong connection between neuron parity score and fairness. This motivates our method in Sec. 5 that conduct alignments the decision rationales of different subgroups explicitly. To avoid confusion, we rewrite the last paragraph of section 3 to avoid potential misunderstanding.

---

> ### Author Response · Authors · 2022-11-23
> **Message from Authors**
>
> Since we're still in the second stage of discussion, we'd like to know what the reviewer thinks of our revision. Has our response helped the reviewer address the reviewer's concerns and reconsider the value of our work? We still have time to discuss if there are any new issues, and we'll be pleased to address them.
>
> We are looking forward to hearing from the reviewer.
>
> Thanks.

---

> > ### Comment · Reviewer_HMGh · 2022-12-01
> > **Late reply**
> >
> > Thank you for the response, and the reminder. I still have some unresolved issues regarding the motivation of using "neuron parity score" for fairness.
> >
> > Thank you for the pointer to the materials in section 4.2. Now I do think that authors are performing "decision rationale analysis" but still I do not think it fully justifies why optimizing "neuron parity score" is important. It is true that "neuron parity score" has some correlation with DP whenever the samples are generated by changing the regularization intensity of FairReg, but why would it be a necessarily better regularizer than the original FairReg regularizer? There may be countless other quantities with positive correlation with DP---but why the neuron parity score, specifically? Is the correlation coefficient particularly large, for instance? In other words, I think the observation 1 in sec 4.2. saying that the neuron parity score is "highly correlated" is vacuous, as we do not get any answer to "higher than what?"
> >
> > Another thing that is missing is whether a "direct regularization" of the neuron parity score indeed gives a better tradeoff between the neuron parity score and the performance.

---

> > > ### Author Response · Authors · 2022-12-04
> > > **Response to Reviewer HMGh (2/2)**
> > >
> > > **Another thing that is missing is whether a "direct regularization" of the neuron parity score indeed gives a better tradeoff between the neuron parity score and the performance.**
> > >
> > > Thanks for the question. We argue that a "direct regularization" indeed decreases the neuron parity score while enhancing the fairness of networks under similar accuracies. Actually, we have split the above question into two sub-questions in our submission:
> > >
> > > SubQ1: Whether the “direct regularization” could help enhance the fairness of networks?
> > >
> > > SubQ2: Whether the “direct regularization” could reduce the decision rationale misalignment across subgroups?
> > >
> > > Note that, we answer the two questions with the same experiment setups and the same baseline methods. Hence, the results of the two sub-questions can definitely infer the mentioned conclusion. We will add a direct analysis of the relationship between neuron parity scores and fairness scores when using the proposed regularization method since the submission could not be further revised at this stage.
> > >
> > > For the SubQ1, we have reported the results in Fig 3 where we show that our DRAlign consistently improves the fairness of the FairReg(\*, noAug) method. Note that, compared with FairReg(\*, noAug), DRAlign has an extra term, i.e., the proposed “direct regularization” term. As we reported in our paper, on the Adult dataset, DRAlign has a relative 67% DP improvement over the FairReg(∆DP, noAug) when both get around 0.770 AP (i.e. the -DP score increases from -0.084 to -0.028). On the CelebA dataset, when the AP values of DRAlign and FairReg(∆DP, noAug) arrive at 0.840, the -DP values of the two methods are around -0.270 and -0.180. These experimental results show the “direct regularization” could help enhance the fairness of networks saliently and sacrifice less accuracy.
> > >
> > > For the SubQ2, we have reported the results in Fig 4. The decision rationale similarity is the approximation of the decision rationale alignment degree via $\sum_{l=0}^L cos(c_l^{a=0}, c_{l}^{a=1}) $. The higher decision rationale similarity means the intermediate neurons have similar responses to different subgroups. Comparing the original FairReg(∆DP, noAug) with our method (i.e., the maximum similarity line) in Fig4 for the large-scale network (i.e., c200) under different lambda setups (i.e., the solid purple line vs. the solid black line),  we can find that DRAlign always gets higher decision rationale similarity (i.e.,  lower decision rationale misalignment / smaller neuron parity score) than the original regularization method FairReg(∆DP, noAug). These results could demonstrate that the “direct regularization” can reduce the decisional rationale misalignment across subgroups. [We here further clarify the Maximum Similarity for easier understanding. Our approximation to the neuron parity score — decision rationale similarity ($\sum_{l=0}^L cos(c_l^{a=0}, c_{l}^{a=1}) $) provides definite upper bounds to the alignment degree. For example, when all layers are aligned for the 3-layer MLP, the decision rationale similarity for each layer is 1 and the decision rationale similarity for the whole model is 3 (Maximum similarity) as shown in Figure 4. Sorry for the unclear depiction, we will add a neuron parity score results report section to our future version.]

---

> > > ### Author Response · Authors · 2022-12-04
> > > **Response to Reviewer HMGh (1/2)**
> > >
> > > **Regarding the motivation of using "neuron parity score" for fairness.**
> > >
> > > Thanks for further clarifying the concerns. We answer the concerns from the following aspects:
> > >
> > > 1. **Neuron parity score has its special meaning.** We define the neuron parity score for each parameter of a network, which measures whether the parameter has consistent responses to different subgroups (e.g., whether the parameter has consistent responses to male or female groups). As a result, the scores could be used to evaluate the importance of intermediate neurons to the fairness of the whole network. The responses of these intermediate neurons on one subgroup represent the decision rationale of the network on that subgroup.
> > >
> > > 2. **The positive correlation between neuron parity scores and the fairness scores demonstrates that more neurons having similar responses to different subgroups lead to fairer models.** In other words,  higher alignment decision rationales across subgroups could lead to fairer models.
> > >
> > > 3. **It may have other positive-correlated quantities (we do not know what quantities the reviewer specifically refers to) but these quantities could not represent the decision rationale alignment degrees.** Note that, the proposed score is a straightforward way to represent the degree of the decision rationale alignment, and is inspired by the decision process-related works that analyze the neuron behaviors under different subgroups and define the decision rationales for different subgroups.
> > >
> > > 4. **The original FairReg(\*, noAug) is a special case of the proposed decision rationale alignment**. At a high-level understanding, we argue that the original FairReg forces the final outputs to have similar responses to different subgroups and can be regarded as a special case of the proposed decision rationale alignment where only the final outputs are considered and the responses of intermediate neurons are neglected. In contrast, our method jointly considers the final outputs and intermediate neurons. As a result, our method can naturally outperform existing FairReg(\*, noAug) methods.
> > >
> > > 5. Moreover, we can check the DP results on the Celeba dataset in Figure 3 (the results are averaged over 10 runs). When the AP values of DRAlign and FairReg arrive at 0.84, the -DP values of the two methods are around -0.27 and -0.18, which shows that the improvement brought by decision rationale alignment is salient. From the experimental results, we can also find that the fairness improvement by minimizing the neuron parity score is important.

---

### Official Review · Reviewer_3JWw · 2022-10-25

**Confidence:** 4
**Correctness:** 4
**Technical Novelty And Significance:** 3
**Empirical Novelty And Significance:** 3
**Recommendation:** 5

**Clarity, Quality, Novelty And Reproducibility:**

Clarity:
It is valuable to study the relationship between alignment and fairness penalty function. It is clear that the neuron alignment combined with the fairness penalty function help achieving fairness in a thorough way. I have a question that when if there is no penalty function for fairness, the alignment can affect improving fairness. At first glance, the alignment itself cannot affect fairness. The more interesting part is that alignment cannot be biased for the DP or EO. The basic concept of alignment seems related to the DP. Therefore, it can be an interesting point to clarify the property of alignment related to other fairness metrics, such as predictive parity, counterfactual fairness, etc.

Quality:
Well-written. The editing of the paper can be improved. In figures, the fonts of legends are somewhat small, and the use of $\Delta DP, no AuG$ and $\Delta DP, Aug$ are inconsistent. Also, the "combined with human society" Section does not provide much insight or philosophy.

Novelty and reproducibility:
The approach and algorithm are somewhat novel, which is valuable. The reproducibility does not have a severe problem because the authors used a well-known algorithm with regularizes.



**Details Of Ethics Concerns:**

This paper studies the fair algorithm, which can have a social impact.

**Strength And Weaknesses:**

Pros:
This paper proposed a new way to achieve fairness by adjusting the inner learning neurons. This approach is interesting and sheds a way to achieve fairness thoroughly. Experiments were conducted, combined with DP and EO. Intensive

Cons:
More analytics analyses such as the relationship between alignment and various fairness metrics, are not thoroughly studied. In the experiments of Credit, FairReg (Aug) shows better performance in DP. It raises the issue of how augmentation can help the DRAlign algorithm. However, it was not addressed.


**Summary Of The Paper:**

The motivation of this paper is to achieve fairness by modifying the learning of the neuron of deep networks (addressing decision rationale misalignment). The proposed algorithm is feasible for large weights in deep networks, and experiments results show that the algorithm makes the result more fair, combined with a conventional regularizer for fairness (measured by DP, EO, etc.) Analysis shows the data augmentation can help achieve fairness in some datasets.


**Summary Of The Review:**

This paper considers an interesting approach with alignment, which can make the fairness algorithm plenty. Experiment results seem promising. The deficit is not thoroughly studied with various fairness metrics.

---

> ### Author Response · Authors · 2022-11-19
> **Response to Reviewer 3JWw (2/2)**
>
> 3. **I have a question that when if there is no penalty function for fairness, the alignment can affect improving fairness. At first glance, the alignment itself cannot affect fairness.**
>
> Thanks for the comments. We find that the alignment itself could still slightly improve fairness when fairness regularization is removed. Specifically, we remove the $L_\text{fair}$  term in Eq.(6) and retain the classification loss and the decision rationale alignment loss. We denote this version as $\text{w.o.FairReg-DRAlign}$. Compared with the model only trained with the classification loss (i.e., $\text{w.o.FairReg}$ ), $\text{w.o.FairReg-DRAlign}$ increase the experimental results (AP, -DP) from (0.776, -0.16 ) to (0.781, -0.14). The results are consistent with our observation that our decision rationale alignment method could further improve fairness and demonstrate that decision rationale alignment is actually a favorable supplement for existing fairness regularization terms. We have added this discussion to Figure 2 and Supplementary Materials A.9.
>
>
> 4. **In figures, the fonts of legends are somewhat small, and the use of  ΔDP,noAuG and ΔDP,Aug are inconsistent. Also, the "combined with human society" Section does not provide much insight or philosophy.**
>
> Thank you. We carefully adjust the fonts of legends including Figure 3 and Figure4. We also revise the “combined with human society” section. Moreover, we have done further proofreading.
>
> &nbsp;
>
> &nbsp;
>
>
> [0] Ching-Yao Chuang and Youssef Mroueh. Fair mixup: Fairness via interpolation. In International Conference on Learning Representations, 2021.
>
> [1] Matt J Kusner, Joshua Loftus, Chris Russell, and Ricardo Silva. Counterfactual fairness. In
> I. Guyon, U. Von Luxburg, S. Bengio, H. Wallach, R. Fergus, S. Vishwanathan, and R. Gar-
> nett (eds.), Advances in Neural Information Processing Systems, volume 30. Curran Asso-
> ciates, Inc., 2017.
>
> [2] P. Garg, J. Villasenor, and V. Foggo. Fairness metrics: A comparative analysis. In 2020 IEEE
> International Conference on Big Data (Big Data), pp. 3662–3666, Los Alamitos, CA, USA,
> dec 2020. IEEE Computer Society

---

> > ### Comment · Reviewer_3JWw · 2022-11-21
> > **Reply to response**
> >
> > Thanks for your reply. In your report, the main term for fairness is the regularization of the fairness metric. In this case, your finding is limited because decision rationale alignment cannot be an independent measure or algorithm. In this case, your algorithm should be examined in various fitness metrics, algorithms, and intensive datasets.

---

> > > ### Author Response · Authors · 2022-11-21
> > > **Response to Reviewer 3JWw**
> > >
> > > Thanks for your prompt reply.  Our work proposes to think the fairness from the decision-making process instead of the final outputs and proposes a score to measure the importance of each neuron to the final fairness score. This is not explored yet in the community.
> > >
> > > **Note that, we never aim to deny the existing regularization methods but to find ways to further enhance them. In a high-level understanding, we argue that the existing regularization terms we use are a part of the decision rationale alignment since the final output is a part of the decision process.** Here, we explicitly split the final output alignment and the intermediate neuron alignment into two parts (i.e., regularization terms and the proposed term) to make the readers understand the main differences between our work and previous works easily.
> > >
> > > **It is unreasonable to just compare our proposed term without fairness regularization terms to existing methods. More specifically, existing regularization methods could be regarded as a special case of the decision rationale alignment where only the final output is considered for alignment.**  From this angle, we see that adding fairness constraints to the final outputs (i.e., existing regularization methods) can align the responses of the intermediate neurons in an implicit way but do not consider the constraints on the intermediate neurons straightforwardly. Our method is designed to fill the gap. The results have totally validated the effectiveness of our method: adding the intermediate neuron alignment can further enhance the fairness even when we remove the existing regularization terms.  **If we compare our term without the fairness regularizations with existing methods, it means that we only do the alignments for the intermediate neurons, which is also unreasonable.**  Moreover, our work might open a door to inspire more explainable ideas for the fairness. Thanks for the insightful question. We will add this discussion to our future version.
> > >
> > > In terms of the experiments, we have done our best to cover baselines and datasets in the community to validate the effectiveness of our method.  Moreover, unlike other fairness-relevant works, our work reports the full tradeoff curve (which is thorough and comprehensive), instead of showing only a single data point which is more adopted in the community like in [3][4]. Moreover, we have tried our best to compare with other fairness works, which could be evaluated by both the DP and EO metric. Still, we will explore the decision rationale under other fairness objectives and compare our methods with baseline methods which could be used to improve the corresponding fairness objectives in the future.
> > >
> > > &nbsp;
> > >
> > > &nbsp;
> > >
> > > [0] Ching-Yao Chuang and Youssef Mroueh. Fair mixup: Fairness via interpolation. In International Conference on Learning Representations, 2021.
> > >
> > > [1] Matt J Kusner, Joshua Loftus, Chris Russell, and Ricardo Silva. Counterfactual fairness. In I. Guyon, U. Von Luxburg, S. Bengio, H. Wallach, R. Fergus, S. Vishwanathan, and R. Garnett (eds.), Advances in Neural Information Processing Systems, volume 30. Curran Associates, Inc., 2017.
> > >
> > > [2] P. Garg, J. Villasenor, and V. Foggo. Fairness metrics: A comparative analysis. In 2020 IEEE International Conference on Big Data (Big Data), pp. 3662–3666, Los Alamitos, CA, USA, dec 2020. IEEE Computer Society.
> > >
> > > [3] Ramaswamy, Vikram & Kim, Sunnie & Russakovsky, Olga. (2021). Fair Attribute Classification through Latent Space De-biasing. 9297-9306. 10.1109/CVPR46437.2021.00918.
> > >
> > > [4] Z. Wang, K. Qinami, I. Karakozis, K. Genova, P. Nair, K. Hata, and O. Russakovsky. Towards fairness in visual recognition: Effective strategies for bias mitigation. In 2020 IEEE/CVF Conference on Computer Vision and Pattern Recognition (CVPR), pp. 8916–8925, Los Alamitos, CA, USA, jun 2020. IEEE Computer Society. doi: 10.1109/CVPR42600.2020.00894.

---

> > > > ### Comment · Reviewer_3JWw · 2022-11-21
> > > > **Reply to repsonse**
> > > >
> > > > I carefully read the abstract, the abstract seems a proposal for a new fairness standard. Anyway, if your proposal should be accommodated with various fairness metrics, the experiments are too small in various fairness metrics.

---

> > > > > ### Author Response · Authors · 2022-11-23
> > > > > **Response to Reviewer 3JWw**
> > > > >
> > > > > **I carefully read the abstract, the abstract seems a proposal for a new fairness standard. Anyway, if your proposal should be accommodated with various fairness metrics, the experiments are too small in various fairness metrics.**
> > > > >
> > > > > Thanks for further clarifying your concerns. To address your concerns, we mainly conduct the following efforts:
> > > > >
> > > > > (1) Demographic parity (DP) and equalized odds (EO) are the common metrics used by state-of-the-art methods for comparison, and other metrics are rarely used.  Note that, the mentioned counterfactual fairness metric is an individual-level definition [0] that captures the intuition that a decision is fair towards an individual. Our work is to enhance the group fairness and the counterfactual fairness metric cannot be used straightforwardly. We have tried our best to recheck published group-fairness-related papers in recent two years on the top-tier conferences and journals (i.e., NeurIPS, ECCV, CVPR, ICML, ICLR, TPAMI) to assess the popularity of these metrics (See detailed statistics on the link https://anonymous.4open.science/r/fairer_submission-F176/). Clearly, DP and EO are the common metrics used by state-of-the-art methods, and the Equality of opportunity (EOP) and predictive parity (PP)  are the third and fourth popular metrics only adopted by 9 methods and one method, respectively. DP is independent of the ground truth labels, which makes it especially useful when reliable ground truth informaiton is not available. EO considers that different groups could have different distribution in terms of different categories. These characteristics make DP and EO representative and popular.
> > > > >
> > > > > (2) To address the concerns on the scalability of our method, we further evaluate and compare our method with FairReg methods on the third popular metric EOP, and predictive parity (PP). Specifically, we adopt the EOP definition in [1], and the PP definition in [2].
> > > > >
> > > > > $$EOP = TPR_{a=0} / TPR_{a=1} = P(\hat{y} =1| a=0, y=1) /  P(\hat{y}=1|a=1,y=1)$$
> > > > >
> > > > > $$PP = | p(y = 1|a = 0, \hat{y} = 1) - p(y = 1|a = 1, \hat{y} = 1) |.$$
> > > > >
> > > > > Under the above definition, EOP close to 1 and PP close to 0 indicate fair classification results. We carefully modify the FairReg method for the EOP and PP metrics.
> > > > >
> > > > > $$
> > > > > L_{fair, EOP} = \Delta EOP(F) =  E_{\mathbf{x} \sim P_0^1}(F(\mathbf{x})) - E_{\mathbf{x} \sim P_1^1}(F(\mathbf{x})),
> > > > > $$
> > > > >
> > > > > $$
> > > > > L_{fair, PP} =  \Delta PP(F) =  ( E_{\mathbf{x} \sim P_0^0}(F(\mathbf{x})) * N_0^0) / ( E_{\mathbf{x} \sim P_0^1}(F(\mathbf{x})) * N_0^1)   - ( E_{\mathbf{x} \sim P_1^0}(F(\mathbf{x})) * N_1^0) / ( E_{\mathbf{x} \sim P_1^1}(F(\mathbf{x})) * N_1^1),
> > > > > $$
> > > > >
> > > > > $N_0^0$, $N_0^1$, $N_1^0$, $N_1^1$ are the sample numbers of subgroups $D_{00}$, $D_{01}$, $D_{10}$, and $D_{11}$, which satisfy the following attribute and category conditions:  {a=0,y=0}, {a=0,y=1},{a=1,y=0},{a=1,y=1} in the batch of data. The sampling methods of $FairReg(\Delta EOP, noAug)$ and $FairReg(\Delta PP, noAug)$ follow those of $FairReg(\Delta EO, noAug)$ and $FairReg(\Delta DP, noAug)$ respectively. For the EOP metric, we align the decision rationales between subgroups {a=0,y=1} and {a=1,y=1}. For the PP metric, we align the decision rationales we align the decision rationales of ($D_{00}$, $D_{10}$) and the decision rationales of ($D_{01}$,$D_{011}$).
> > > > >
> > > > > We showcase the experimental results here (the report averages over 10 times). From the table, we can see that our method DRAlign consistently improve the fairness performance under the EOP and PP metrics,  that is, our method could be extended to EOP and PP. We will add the above results to our future version.
> > > > >
> > > > > Table A
> > > > >
> > > > > |EOP (β=λ/10)|λ = 0.5|λ = 0.9|λ = 1.0|
> > > > > |:----|:----|:----|:----|
> > > > > |$AP_{DRAlign}$|**0.7820**|**0.7811**|**0.7805**|
> > > > > |$EOP_{DRAlign}$|**0.9636**|**0.9706**|**0.9689**|
> > > > > |$AP_{FairReg(ΔEOP, noAug)}$|0.7813|0.7805|0.7802|
> > > > > |$EOP_{FairReg(ΔEOP, noAug)}$|0.9456|0.9622|0.965|
> > > > >
> > > > > Table B (we set β as 0.1 when λ = 0.0)
> > > > >
> > > > > |PP |λ = 0.0|λ = 0.2|λ = 0.4|λ = 8.0|
> > > > > |:----|:----|:----|:----|:----|
> > > > > |$AP_{DRAlign}$|**0.787**|**0.785**|**0.782**|**0.771**|
> > > > > |$EOP_{DRAlign}$|**0.035**|**0.025**|**0.033**|**0.019**|
> > > > > | |λ = 0.0|λ = 0.8|λ = 6.0|λ = 8.0|
> > > > > |$AP_{FairReg(ΔPP, noAug)}$|0.784|0.7787|0.771|0.768|
> > > > > |$EOP_{FairReg(ΔPP, noAug)}$|0.044|0.038|0.0234|0.021|
> > > > >
> > > > > [0] Matt J Kusner, Joshua Loftus, Chris Russell, and Ricardo Silva. Counterfactual fairness. In I. Guyon, U. Von Luxburg, S. Bengio, H. Wallach, R. Fergus, S. Vishwanathan, and R. Gar- nett (eds.), Advances in Neural Information Processing Systems, volume 30. Curran Asso- ciates, Inc., 2017.
> > > > >
> > > > > [1] Wang, G., Du, M., Liu, N., Zou, N., and Hu, X., “Mitigating Algorithmic Bias with Limited Annotations”, TPAMI, 2022.
> > > > >
> > > > > [2] P. Garg, J. Villasenor, and V. Foggo. Fairness metrics: A comparative analysis. In 2020 IEEE International Conference on Big Data (Big Data), pp. 3662–3666, Los Alamitos, CA, USA, dec 2020. IEEE Computer Society

---

> > > > > > ### Comment · Reviewer_3JWw · 2022-11-24
> > > > > > **Response to reply**
> > > > > >
> > > > > > Thanks for your reply and efforts. If possible, I want to see the results of a simple (one-side or two-side) t.test using 10 times replications to show the superiority of  DRAlign. The gains look small in Table A, and Table B looks strange.

---

> > > > > > > ### Author Response · Authors · 2022-11-25
> > > > > > > **Response to Reviewer 3JWw**
> > > > > > >
> > > > > > > Thanks for your meaningful suggestions. Note that, it is meaningless to conduct the t-test for the average precision ($AP$) or the fairness metric (i.e., $EOP$ or $PP$) alone since we want to maintain high fairness score and high $AP$ at the same time. Hence,  we try to consider both the average precision ($AP$) and the fairness metrics (i.e., $EOP$ or $PP$) for t-test. To this end, for the evaluation result at a time, we have the $AP$ score and the fairness metric score (i.e., $EOP$ or $PP$). Meanwhile, we also get the minimum $AP$ score and fairness metric score across all compared results, which are denoted as the  $AP_{min}$ and  $EOP_{min}$ or $PP_{min}$. Then, we multiply the $(AP-AP_{min})$ with the fairness metric score $(EOP-EOP_{min})$ or $(PP-PP_{min})$,  and get a comprehensive score, i.e, the rectangle area bounded by the two points ($AP$, $EOP$ or $PP$) and ($AP_{min}$, $EOP_{min}$ or $PP_{min}$). We denote them as $RectArea_{AP, EOP}$ and $RectArea_{AP, PP}$, respectively. Note that, the larger $RectArea_{AP, EOP\,or\,PP}$ means fairer under similar accuracy via the $EOP$ metric or $PP$ metric.
> > > > > > >
> > > > > > > Then, we conduct the t-test on the $RectArea_{AP, EOP}$ and $RectArea_{AP, PP}$, respectively. From the two tables, we can see that under most parameter settings, the p-values < 0.02. This means that our results are statistically significant. However, when the λ is large, the p-value is also large. We think this is because the decision rationale alignment degree rises when λ increases to a high value as we claimed in our work, which makes our improvement not that salient when compared with the impact of different random seeds.
> > > > > > >
> > > > > > > We do appreciate your suggestions to further highlight our advantages. We have several points to reclaim.
> > > > > > > We think the improvement gains are not small. Taking $EOP$ as an example, when λ is set as 0.5, we see that the $EOP$ value increases from 0.9456/1.0 to 0.9636/1.0 as shown in our last response. The gap to the highest score (1.0) decreases from 0.0534 to 0.0364, which improves 32%. We can also check the $DP$ results on the CelebA dataset in Figure 3. When the $AP$ values of DRAlign and FairReg($\Delta DP$, noAug) arrive at 0.84, the $-DP$ values of the two methods are around -0.27 and -0.18, which shows that the improvement is salient.
> > > > > > >
> > > > > > > |$RectArea_{AP, EOP}$|λ =0.5|λ = 0.9|λ = 1.0|
> > > > > > > |:---:|:---:|:---:|:---:|
> > > > > > > |p_value|0.008|0.018|0.111|
> > > > > > >
> > > > > > > |$RectArea_{AP, PP}$|λ = 0.0|λ = 0.2|λ = 0.4|λ = 8.0|
> > > > > > > |:---:|:---:|:---:|:---:|:---:|
> > > > > > > |p_value|0.019|0.001|0.0028|0.227|

---

> > > > > > > > ### Comment · Reviewer_3JWw · 2022-11-25
> > > > > > > > **Reply to response**
> > > > > > > >
> > > > > > > > I am disappointed that the authors do not answer my question. My question is merely about the t.test of EOP or PP, not concerning the AP. The cocept of Rect Area is not introduced in the paper, and it have no reference.

---

> > > > > > > > > ### Author Response · Authors · 2022-11-26
> > > > > > > > > **Respnse to Reviewer 3JWw**
> > > > > > > > >
> > > > > > > > > Thanks for your reply. We thought you wanted us to highlight the advantages of our method according to your last response. Here we merely show the results of t-test for EOP and PP metrics.
> > > > > > > > >
> > > > > > > > > |EOP|λ = 0.5|λ = 0.9|λ = 1.0|
> > > > > > > > > |:----|:----|:----|:----|
> > > > > > > > > |p value|0.002|0.134|0.335|
> > > > > > > > >
> > > > > > > > > |PP|λ = 0.0|λ = 0.2|λ = 1.0|λ = 8.0|
> > > > > > > > > |:----|:----|:----|:----|:----|
> > > > > > > > > |p value|0.078|0.058|0.088|0.320|

---

> > > > > > > > > > ### Comment · Reviewer_3JWw · 2022-11-27
> > > > > > > > > > **Reply to Response**
> > > > > > > > > >
> > > > > > > > > > Thanks for your reply. With a significant level of 0.05, only EOP with $\lambda=0.5$ is significant. Totally, the gains seem relatively small in this case. I cannot be positive.

---

> > > > > > > > > > > ### Author Response · Authors · 2022-11-29
> > > > > > > > > > > **Respnse to Reviewer 3JWw**
> > > > > > > > > > >
> > > > > > > > > > > Thanks for the summary. As we have mentioned in the response on 26 Nov 2022, fairness enhancement methods should consider both the average precision (AP) and fairness metric value, that is, the desired fairness method should enhance the model’s fairness metric while having little influence on the AP.
> > > > > > > > > > >
> > > > > > > > > > > In our submission, we use a 2D plot to achieve this where the x-axis and y-axis denote the fairness metric and the AP, respectively (See Figure 2 and 3). In terms of the t-test, when we only evaluate the EOP or PP, the advantages are not significant. However, when we consider the AP and the Fairness metric jointly, the advantages are significant (See the response on 26 Nov 2022). We hope the reviewer could consider this during the evaluation. Thanks.

---

> ### Author Response · Authors · 2022-11-19
> **Response to Reviewer 3JWw (1/2)**
>
> Thank you for the constructive comments and advice. Our detailed responses are shown below:
>
> 1. **More analytics analyses such as the relationship between alignment and various fairness metrics are not thoroughly studied.**
>
> Thanks for your kind advice. There are several fairness metrics used by existing works, i.e., DP, EO, predictive parity, and counterfactual fairness.
> In our submission, we take the DP metric as an example to introduce the alignment method for simplicity and extend our development to EO. Specifically, in the training process, for the DP metric, we randomly sample $D_0$ and $D_1$  subgroups with the same group data size according to the interested attribute from the training data (See line 2 in Algorithm 1).  We characterize the decision rationale for $D_0$  and $D_1$  separately and we align the decision rationales of these two subgroups ($D_0$ , $D_1$ ). Actually, our method can be easily extended to other metrics. For example, when considering the EO metric, we randomly sample four subgroups with the same group size from the training data. The four subgroups are  $D_{00}$ , $D_{01}$ , $D_{10}$ , and $D_{11}$ , which satisfy the following attribute and category conditions: {a=0,y=0}, {a=0,y=1},{a=1,y=0},{a=1,y=1}, respectively. Then, we only align the decision rationales of two subgroups with the same y, that is, we align the decision rationales of ($D_{00}$ , $D_{10}$ ) and the decision rationales of ($D_{01}$ , $D_{11}$ ). We have shown the results in the second row of Figure 3. Our method is also applicable to other fairness metrics that quantify the expected difference between groups. Predictive parity focuses on whether the positive predictive value (PPV) is the same for both groups. We should align the decision rationales for the data in both groups ({a=0, $\widehat{y}$=1} and {a=1,$\widehat{y}$=1}). Counterfactual fairness [1] quantifies fairness from the perspective of an individual [2], which is beyond our current framework. We will further explore it in the future.
>
>
> 2. **In the experiments of Credit, FairReg (Aug) shows better performance in DP. It raises the issue of how augmentation can help the DRAlign algorithm. However, it was not addressed.**
>
>
> Thanks for the insightful comments. The data augmentation and our decision rationale alignment are two different ways to enhance fairness. Intuitively, we can combine the two solutions straightforwardly. For example, we can replace the second term in Eq. (6) (i.e., $L_\text{fair}$) with the data augmentation-embedded term (See [0] for more details) and have a new formulation of Eq. (6).
> $$
> L= E_{(x,y) \sim P}(L_{\text{cls}}(\text{F}(x),y)) + \lambda L_\text{aug}(\text{F}) + \beta \sum_{k=0}^Kd_k,
> $$
> We denote the above method for DP regularization as $\text{DRAlign}(\Delta \text{DP}, \text{Aug})$. We evaluate this version and compare it to the method without augmentation (i.e., $\text{DRAlign}(\Delta \text{DP})$ on the Credit dataset.  We see that: the fairness score (i.e., -DP) increases from -0.0169 to -0.0155 while the average precision (AP) also increases from 0.877 to 0.881, which further demonstrates the scalability of our method. To address the concern, we add a discussion in Supplementary Materials A.8. Please check the revision for details.

---

### Author Response · Authors · 2022-11-19
**Paper Revision Summary**

We appreciate the thoughtful comments and helpful criticism from every reviewer. We are delighted that the reviewers thought our paper was "interesting", "versatile", "easy to follow and clear" and "with interesting analysis".

Below is a summary of our paper update, and we mark the updates in our paper with blue color.

1. **[Introduction]** We add an example to explain "social fairness".
2. **[Related Work]** We introduce more work on fairness including counterfactual fairness. We cite one related work NeuronFair.
3. **[3.3 Limitations]** We rewrite the last paragraph of section 3.3 to clarify the limitation that the accuracy of current methods decreases by a large margin and such methods only provide fairness metric results but neglect the decision-making process.
4. **[4.2 Relationship Between Parity Score And Fairness]** We rewrite the last paragraphs of sections 4.2 to summarize our study and reveal the strong connection between neuron parity score and fairness.
5. **[6.1 Experimental Setup]** We introduce two more baseline methods: adversarial debiasing and a post-processing method.
6. **[6.2 Fairness Improvement Performance]** We rephrase our description and we show results of more baselines.
7. **[Supplementary Materials A.1]** We introduce more about our hyper-parameter setting. Moreover, we introduce how we select the parameter $\beta$.
8. **[Supplementary Materials A.5]** We show the training time estimation.
9. **[Supplementary Materials A.6]** We show the AP values trained on different model architectures as a supplement for [6.3 Connection with over-parameterization].
10. **[Supplementary Materials A.7]** We analyze more fairness metrics as a supplement for the updated content in [3.2 FAIRNESS REGULARIZATION].
11. **[Supplementary Materials A.8]** We analyze the combination of our method with data augmentation as a supplement for the updated content in [6.2 FAIRNESS IMPROVEMENT PERFORMANCE].
12. **[Supplementary Materials A.9]** We analyze our DRAlign method without fairness regularization. The experimental results show that our decision rationale alignment method could still further improve fairness and is a favorable supplement for existing fairness regularization terms.
13. **[Supplementary Materials A.10]** We move the human society analysis into supplementary materials.

---

### Decision · Program_Chairs · 2023-01-20

**Decision:**

Reject

**Justification For Why Not Higher Score:**

Reviewers agree that the paper fails to show the advantages of the proposed algorithm in various metrics without the prediction accuracy. Another common concern from the reviewers is unclear motivation. In particular, reviewers do not see why the proposed “neuron parity score” is necessarily a better regularization objective than other quantities. Although the authors have shown that the proposed “neuron parity score” is positively correlated with fairness, there is no justification on why we should directly minimize “neuron parity score”. Given the possibility of the existence of many more similar metrics other than "neuron parity score", the committee believes that we need a more concrete justification of why “neuron parity score” should matter more than other quantities in order to derive some meaningful conclusions.

**Justification For Why Not Lower Score:**

N/A

**Metareview: Summary, Strengths And Weaknesses:**

This paper proposed a new way to achieve fairness by adjusting the inner learning neurons. The idea is very interesting. However, reviewers agree that the paper fails to show the advantages of the proposed algorithm in various metrics without the prediction accuracy.

Another common concern from the reviewers is unclear motivation. In particular, reviewers do not see why the proposed “neuron parity score” is necessarily a better regularization objective than other quantities. Although the authors have shown that the proposed “neuron parity score” is positively correlated with fairness, there is no justification on why we should directly minimize “neuron parity score”. Given the possibility of the existence of many more similar metrics other than "neuron parity score", the committee believes that we need a more concrete justification of why “neuron parity score” should matter more than other quantities in order to derive some meaningful conclusions.